# E69K mutation in β-tubulin 2 blocks cell wall integrity signaling during plant cell elongation

Huanhuan Yang [ID] [1,2,4], Jie Wang [ID] [3,4], Guangda Wang [ID] [1,4], Chaofeng Wang[1], Xiaxia Zhang[1], Juan Tian[1], Yanjun Yu[1] & Zhaosheng Kong [ID] [1,2,3 ✉]

## Abstract

The intact cell wall is the prerequisite for plant cell morphogenesis. Loss of function in *FRA1/KINESIN-4A*, which encodes a microtubule-based kinesin motor, causes dwarfed growth phenotypes with reduced cell wall mechanics. However, the underlying mechanisms remain elusive. Here, using genetic screening, we identify a suppressor of *fra1* (*sofa1*) mutation that specifically suppresses the dwarf phenotype of the *fra1* mutant. The *sofa1* carries an E69K mutation in β-Tubulin 2 (TUB2), and the dominant suppressive effect of E69K mutation is conserved among β-tubulins. We further reveal that incorporation of TUB2$^{E69K}$ affects microtubule stability, yet fails to rescue the cell wall defects or lateral displacement of microtubules in *fra1*. Combining with transcriptomic analysis, we propose that the E69K mutation of TUB2 potentially restores the cell elongation by blocking the cell wall integrity (CWI) signaling. Our study sheds new light on the complex mechanism underlying the dwarfism of the *fra1* mutant, and further proposes a potential model by which microtubules control plant cell elongation.

**Keywords** Cortical Microtubules; Cell Wall Integrity; Kinesin-4A; Plant Cell Elongation; β-Tubulin
**Subject Categories** Cell Adhesion, Polarity & Cytoskeleton; Plant Biology

## Introduction

The plant cell wall is a complex and dynamic extracellular matrix that surrounds plant cells and plays a crucial role during plant growth and development. The primary cell wall is composed of cellulose, hemicelluloses, pectins, and a small amount of structural proteins, conferring two seemingly incompatible properties on the growing wall: mechanical strength and dynamic extensibility (Cosgrove, 2024). Mechanical strength enables cells to resist turgor pressure and maintain cell morphology, but dynamic extensibility allows cells to irreversibly expand (Coen and Cosgrove, 2023). The

cellulose microfibrils are embedded in a polysaccharide matrix and act as the major load-bearing component of the wall. Each cellulose microfibril, consisting of β-1,4-linked glucan chains, is synthesized at the cell surface by cellulose synthase complexes (CSCs), which move along cortical microtubules (CMTs) (Paredez et al, 2006; Pedersen et al, 2023). By contrast, pectins and hemicelluloses are synthesized in the Golgi apparatus and secreted by vesicle transport (Hoffmann et al, 2021). These polysaccharides form an integrated and complex wall network system by crosslinking reactions, which largely determines plant cell morphogenesis. Accumulative studies have indicated that the cell wall is not a passive physical structure, and it also actively participates in various signaling processes (Delmer et al, 2024). Under abiotic and biotic stresses, cells can timely sense the cell wall damage (CWD) and trigger cell wall integrity (CWI) signaling, activating the expression of stress-related genes and production of defense-related metabolites (Molina et al, 2024; Vaahtera et al, 2019). These adaptive changes lead to a series of compensatory effects, including the cell growth inhibition (Chaudhary et al, 2025; Voxeur and Höfte, 2016).

Fragile Fiber1 (FRA1) / KINESIN-4A, a member of the Kinesin subfamily of microtubule-based motor proteins, has been shown to move processively along CMTs (Zhong et al, 2002; Zhu and Dixit, 2011). Accumulating evidence suggests that FRA1 transports secretory vesicles containing non-cellulosic components to exocytic sites and regulates cell wall mechanics during plant cell morphogenesis (Kong et al, 2015; Zhu et al, 2015). Although FRA1 does not directly alter microtubule organization and the motility of CSCs (Kong et al, 2015), a recent study has demonstrated that FRA1 modulates the lateral displacement of CMTs through interaction with cellulose synthase-microtubule uncoupling (CMU)/kinesin light chain-related (KLCR) proteins (Ganguly et al, 2020), indicating its multifaceted roles beyond vesicle transport. Loss of function of *FRA1* causes a dramatic reduction in mechanical strength and a marked dwarf phenotype, characterized by significantly shorter cell lengths than wild-type (WT) plants (Zhong et al, 2002). In this study, we identified an E69K mutation in TUB2 that suppresses the dwarf phenotype of the *fra1* mutant. This mutation is a gain-of-function mutation, which alters CMTs stability and potentially impacts cell wall integrity signaling. Our findings reveal a potential genetic mechanism of FRA1 and microtubules in cell elongation control and provide a promising site on β-tubulins for governing plant cell growth.

[1]Department of Agri-microbiomics and Biotechnology, State Key Laboratory of Microbial Diversity and Innovative Utilization, Institute of Microbiology, Chinese Academy of Sciences, Beijing, China. [2]University of Chinese Academy of Sciences, 100049 Beijing, China. [3]Houji Laboratory in Shanxi Province, Shanxi Agricultural University, 030801 Taigu, China. [4]These authors contributed equally: Huanhuan Yang, Jie Wang, Guangda Wang. ✉E-mail: zskong@im.ac.cn

# Results and discussion

## The E69K mutation in TUB2 suppresses the dwarf phenotype of *fra1*

By genetic screening, we isolated a suppressor of fra1 (designated as *sofa1*) with a significant increase in plant height (Fig. 1A,B). The anatomical examination showed longer pith cells in *sofa1* stems (Figs. 1C and EV1A), indicating that the restored plant height in the *sofa1* plants is caused by the restoration of cell expansion. Moreover, *sofa1* plants produce longer leaves and siliques, comparable to the WT control (Fig. EV1B–D). These results indicated that the organ elongation defects of *fra1* are suppressed overall in the *sofa1* mutant. The *sofa1* mutant was backcrossed with *fra1* to generate the BC1F1 generation, and all BC1F1 progeny showed indistinguishable plant height from *sofa1* plants (*sofa1*-like), indicating that *sofa1* might be a dominant mutation. BC1F2 generation obtained by selfing F1 plants segregated, and the segregation ratio of the two phenotypes (*fra1*-like and *sofa1*-like) was close to 1:3 (142:425; $\chi^2 = 0.96 < \chi^2_{(0.05)} = 3.84$). Thus, the genetic analysis showed that *sofa1* is a dominant mutant caused by a single gene mutation.

Next, we isolated the causative gene using a MutMap method (Takagi et al, 2015). Based on the whole-genome sequencing, we identified a G-A substitution at the 205th base of the *TUB2* (*At5g20690*) gene on chromosome 5 (Fig. 1D), resulting in a missense mutation of the acidic Glu69 to the basic Lys69 (E69K) (Fig. 1E). We found the G205A mutated *TUB2* (encoding TUB2$^{E69K}$) genomic fragment significantly increased the plant height in *fra1* (Fig. EV1E,F). These results confirmed that the E69K mutation in TUB2 is responsible for the *sofa1* and suppresses the dwarf phenotype of *fra1*.

## The dominant effect of E69K mutation is conserved among β-tubulins

TUB2 is one of the nine β-tubulin (TUB) proteins in Arabidopsis, which are the basic building blocks of microtubules (MTs) together with α-tubulin (TUA) (Nogales et al, 1998). The protein sequence alignment showed that Glu69 is highly conserved in various β-tubulins (Fig. 1F). Since several tubulin genes function redundantly in a range of tissues, the null mutations of a single tubulin gene usually exhibit no obvious phenotype (Hashimoto, 2013). To confirm that the E69K mutation of TUB2 is a gain-of-function mutation rather than a null mutation, we mutated the *TUB2* gene by the CRISPR-Cas9 technology and obtained the *TUB2* knockout mutants in *fra1* (*fra1 tub2c*) and *sofa1* (*sofa1 tub2c*) (Fig. EV2G).

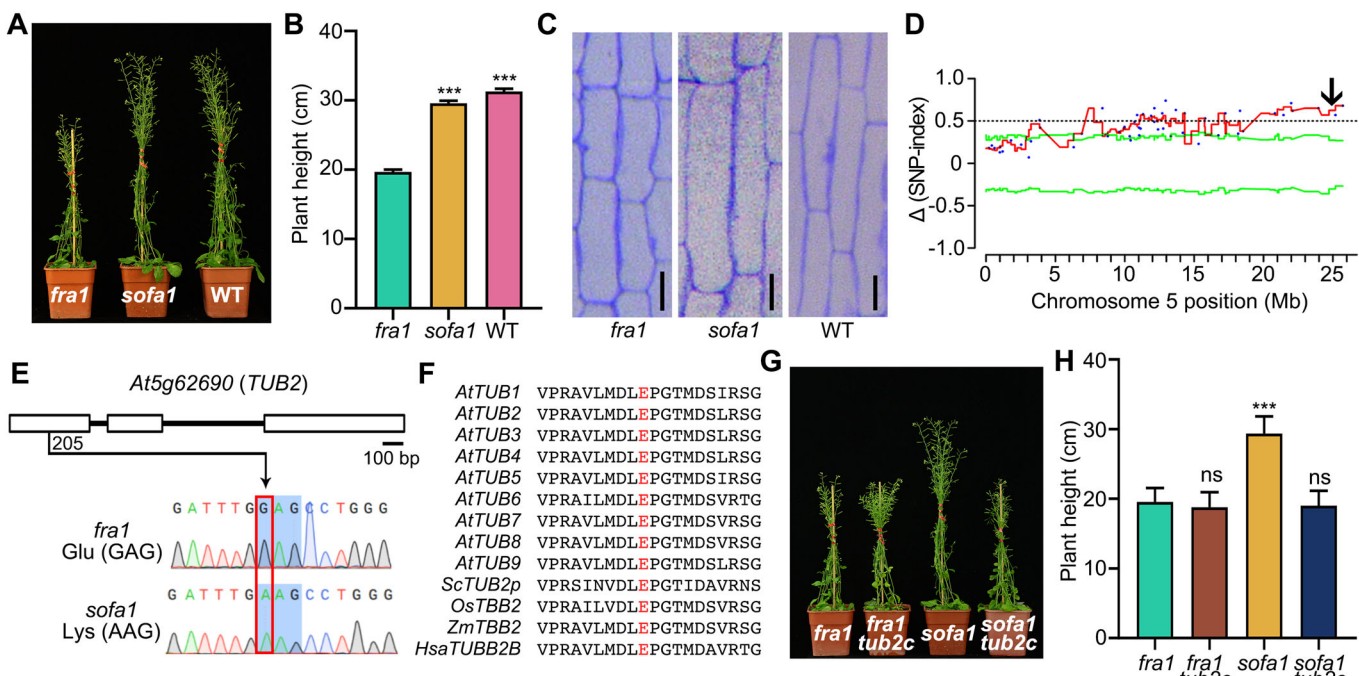

**Figure 1. Identification and gene mapping of the suppressor of *fra1*.**

(A, B) Growth phenotype of *fra1*, *sofa1*, and WT plants (A) and quantification of their plant height (B). More than 20 plants were counted for each sample. (C) Longitudinal sections of pith cells in basal stems of *fra1*, *sofa1*, and WT plants. Scale bars indicate 50 μm. (D) Δ(SNP-index) plot of the chromosome 5 generated by MutMap analysis. Blue dots correspond to SNPs, and the red line represents the sliding window average values of SNP indices of 1-Mb intervals with a 1-kb increment. Green lines show the 95% confidence limit of SNP index value; The black arrow indicates the candidate region for *TUB2*. (E) Structure and sequencing identification of *TUB2* gene. Open boxes represent exons, and black lines denote introns. The red box indicates a G-A substitution at the 205th base of *TUB2*. (F) Multiple sequence alignment of nine TUBs in *Arabidopsis thaliana* (AtTUB1-AtTUB9), *Saccharomyces cerevisiae* TUB2p (ScTUB2p), *Oryza sativa* TBB2 (OsTBB2), *Zea mays* TBB2 (ZmTBB2) and *Homo sapiens* TUBB2B (HsaTUBB2B). The sequences were aligned with MUSCLE, and the conserved Glu69 is highlighted in red. (G, H) Growth phenotype of *fra1*, *fra1 tub2c*, *sofa1*, and *sofa1 tub2c* plants (G) and quantification of their plant height (H). More than 20 plants were counted for each sample. Data information: In (B), data are presented as mean ± SD. ***$P < 0.001$ with *fra1* by one-way ANOVA, $P < 0.0001$ (*sofa1*), $P < 0.0001$ (WT). In (H), data are presented as mean ± SD. ***$P < 0.001$ and ns ($P > 0.05$), no significant difference with *fra1* by one-way ANOVA. $P = 0.5164$ (*fra1 tub2c*), $P < 0.0001$ (*sofa1*), $P = 0.7509$ (*sofa1 tub2c*). Source data are available online for this figure.

The *fra1 tub2c* plants displayed a similar phenotype to *fra1* plants (Fig. 1G,H). However, the plant height of *sofa1 tub2c* returned to that of the *fra1* single mutant (Fig. 1G,H). These results indicated that the E69K mutation of TUB2 is a gain-of-function mutation, explaining why a point mutation in one of the nine β-tubulins would strongly suppress the dwarf phenotype of *fra1*. Moreover, we transformed a mutated *TUB6* construct encoding the TUB6[E69K] into *fra1*, and the transgenic lines also showed an increase in plant height (Fig. EV1H,I). These results demonstrated that the TUB2 mutation from Glu69 to Lys69 is a gain-of-function mutation, and represents a conserved regulatory mechanism among several β-tubulins.

## The 69[th] residue of TUB2 plays a key role in plant height control of *fra1*

It is known that each β-tubulin binds a molecule of GTP, where GTP hydrolysis and nucleotide exchange occur (called E-site) (Howard and Hyman, 2003; Manka and Moores, 2018). At the E-site, $Mg^{2+}$ is tightly linked to GTP binding and is essential for tubulin polymerization and GTP hydrolysis (Bhattacharya et al, 1994; Fees and Moore, 2018; Grover and Hamel, 1994). We analyzed the resolved crystal structure of yeast β-tubulin (Ayaz et al, 2012) and found that the conserved Glu69 residue resides in the T2 loop, which is localized in the entrance of the pockets that allow the tight binding of GTP and the $Mg^{2+}$ (Fig. EV2A). The direct contact between the acidic Glu69 and the magnesium ion leads us to hypothesize that the positively charged amino acid in the 69[th] residue may affect the coordination of the magnesium ion and thus the plant height of *fra1*. We mutated the Glu69 in TUB2 to other positively charged amino acids, arginine (69R) and histidine (69H), and transformed them into *fra1*. As expected, all positively charged amino acid at the 69[th] residue of TUB2 suppressed the dwarf phenotype of *fra1* (Fig. EV2B,C). These results suggested that the charge change at the 69[th] amino acid of TUB2 plays a critical role in regulating plant height in the *fra1* background.

Given that the E69K mutation in TUB2 was identified in a FRA1-deficient background, we asked whether there were any genetic relationships between *TUB2* and *FRA1* in plant height control. To address this question, we identified the *tub2[E69K]* point mutant from the F2 population of a cross between *sofa1* and Col-0, which contains an E69K mutation in TUB2 and removed FRA1-deficient background. Interestingly, the *tub2[E69K]* mutants did not exhibit increased plant height compared to WT plants (Fig. EV2D,E). These results indicated that the TUB2[E69K]-mediated restoration of plant height is a non-additive effect and dependent on the FRA1-deficient background.

## The incorporation of TUB2[E69K] influences the stability of microtubules

Next, we generated pTUB2::VisGreen-TUB2 and pTUB2::VisGreen-TUB2[E69K] constructs and introduced them into *fra1* to visualize CMTs directly. The arrangement of CMTs is generally observed to be perpendicular to the growth axis in fast-elongating cells (Lloyd and Chan, 2004). As shown in Fig. 2A, the ordered microtubules were clearly observed, either labeled by VisGreen-TUB2 or VisGreen-TUB2[E69K]. Quantitative measurement of the fluorescence intensity

showed that the amount of TUB2[E69K] was significantly lower than that of TUB2 (Fig. 2B), indicating TUB2[E69K] could be uniformly incorporated into CMTs, but in smaller amounts than wild-type TUB2. We further determined whether the incorporation of TUB2[E69K] affects dynamic properties of microtubules. Notably, although the shrinkage rate, catastrophe frequency, and rescue frequencies are not markedly different among these transgenic lines (Fig. 2D–F), the incorporation of TUB2[E69K] significantly reduces the microtubule growth rate (Fig. 2C).

The microtubule-disrupting drug oryzalin binds to tubulin, making extant microtubules more likely to depolymerize (Nakamura et al, 2004). To further explore the effects of TUB2[E69K] on the stability of microtubules, we treated 6-day-old seedlings with a series of concentrations of oryzalin and found CMT arrays disappeared relatively faster in *fra1* + *TUB2[E69K]* petiole cells than in petiole cells of *fra1* + *TUB2* (Fig. 2G,H). Consistent with this result, when plants were grown vertically on the medium containing the oryzalin, *sofa1* roots began swelling at 30 nM oryzalin, compared to 80 nM for the *fra1* (Fig. EV3A–C). Taken together, these results indicated that incorporation of TUB2[E69K] affects the stability of microtubules in the *fra1* mutant background.

## The TUB2[E69K] fails to affect the lateral displacement of microtubules in *fra1*

CMU1/KLCR1 encodes a tetratricopeptide repeat-related protein and has been reported to interact with certain IQD (IQ67-domain) proteins (Abel et al, 2013; Burstenbinder et al, 2013; Zang et al, 2021). Moreover, CMU1 proteins have been shown to prevent lateral displacement of microtubules during cellulose synthesis (Liu et al, 2016). A recent study has reported that the *fra1* mutant showed significantly reduced CMU1 signal along CMTs, leading to severe loss of lateral displacement of CMTs (Ganguly et al, 2020). We introduced mCherry-CMU1 markers into *fra1*, *sofa1*, and WT plants and found that the protein levels of mCherry-CMU1 in *sofa1* and *fra1* were comparable, and both were significantly lower than those in WT plants (Fig. 3A,D). We performed further live-cell observations in WT, *fra1* and *sofa1* plants expressing VisGreen-TUB6. The lateral displacement of CMTs was observed in *sofa1* mutant, which could be seen drifting in the cell cortex (Fig. 3B; Movie EV1). By kymograph analysis, we found that the frequency of lateral microtubule displacements in *sofa1* is significantly higher than that in WT plants, which is indistinguishable from that in *fra1* mutant (Fig. 3C,E). These results suggested that the E69K in TUB2 fails to affect the lateral displacement of microtubules in *fra1*, and the mechanism of TUB2[E69K] mutation on plant height control is disassociated from the lateral displacement of microtubules.

## *Sofa1* mutants still maintain the cell wall defects as in *fra1*

Loss of FRA1 function leads to evident cell wall defects, manifested as brittle stems and altered arabinose content (Kong et al, 2015; Zhu et al, 2015). We measured the cell wall mechanical strength and found that the *sofa1* mutants still maintain the brittle-stem phenotype (Fig. 4A). By analyzing cell wall composition, we found that *sofa1* also has significantly more arabinose in the inflorescence stems, comparable to *fra1* (Fig. 4B). It is an interesting finding since cell elongation and cell wall mechanics have long been considered to be tightly linked. *Sofa1* mutants still maintain the cell wall defects as in *fra1*, suggesting cell

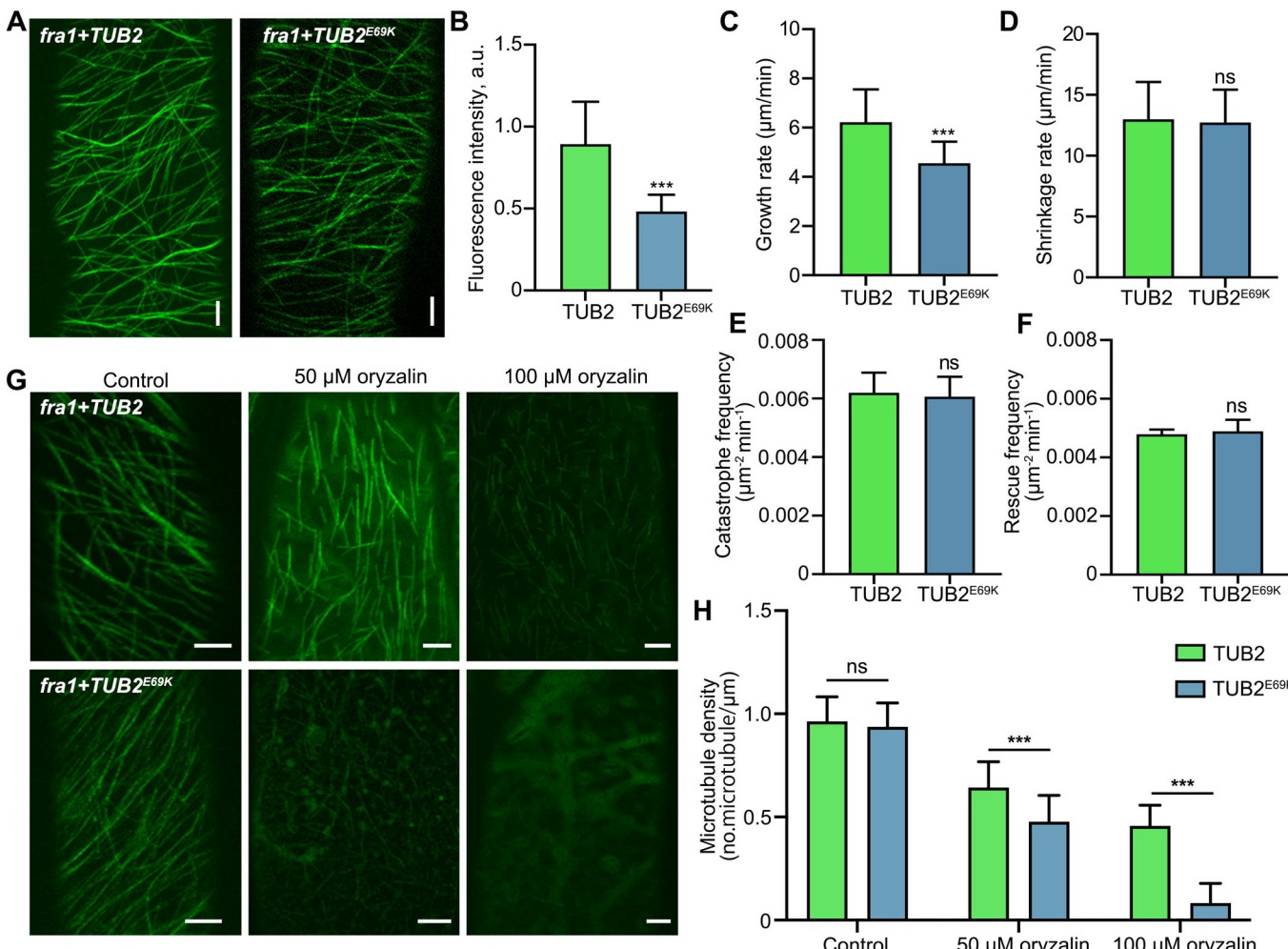

**Figure 2. The incorporation of TUB2^E69K influences the stability of microtubules.**

(A) Cortical microtubules were visualized in *fra1* mutants labeled by VisG-TUB2 (TUB2) and VisG-TUB2^E69K (TUB2^E69K) in the leaf epidermal cells. Scale bars indicate 5 μm. (B) Fluorescence intensity of cortical microtubules labeled by VisG-TUB2 and VisG-TUB2^E69K in *fra1* mutants. *n* = 16 cells from eight individual plants. a.u., arbitrary units. (C–F) Quantification of microtubule growth rate (C), shrinkage rate (D), catastrophe frequency (E), and rescue frequency (F) in *fra1* + *TUB2* and *fra1* + *TUB2^E69K*. More than 200 microtubules from five individual plants were counted. (G) Cortical microtubules of the leaf epidermal cells in *fra1* + *TUB2* and *fra1* + *TUB2^E69K* treated with control, 50 μM oryzalin, 100 μM oryzalin for 24 h. *n* = 5 biological replicates. Scale bars indicate 5 μm. (H) Microtubule density in *fra1* + *TUB2* and *fra1* + *TUB2^E69K* after oryzalin treatment. *n* = 30 cells from five individual plants were counted. Data information: In (B–H), data are presented as mean ± SD. ***$P < 0.001$ and ns ($P > 0.05$), no significant difference with TUB2 by Student's *t* test. (B) $P < 0.0001$. (C) $P < 0.0001$. (D) $P = 0.3969$. (E) $P = 0.7590$. (F) $P = 0.6664$. (H) $P = 0.9456$ (Control), $P < 0.0001$ (50 μM oryzalin), $P < 0.0001$ (100 μM oryzalin). Source data are available online for this figure.

elongation and cell wall mechanics are relatively uncoupled in *sofa1*. Furthermore, we examined the cellulose content of cell wall and the localization of cellulose synthase 3 (CESA3) proteins in *sofa1* mutants, with no abnormalities detected (Fig. 4C,D), indicating no change in cellulose biosynthesis and being consistent with the cell wall of *fra1* mutants. Overall, we demonstrated that the E69K mutation in TUB2 uncouples the cell elongation from cell wall mechanics, and restores cell elongation in *fra1* mutants while still maintaining the cell wall defects.

## The TUB2^E69K blocks the regulation of CWI-related gene expression in *fra1*

Acting as a robust yet flexible barrier, the cell wall shows dynamic growth responses. Under stress conditions, the cell wall perturbations could be well-monitored by plants, thus activating cell wall integrity (CWI) signaling and altering related gene expression, ultimately responding to adverse environments, such as the active inhibition of cell elongation (Chaudhary et al, 2025; Vaahtera et al, 2019; Voxeur and Höfte, 2016). Considering that the *sofa1* mutants still maintain the cell wall defects as in *fra1*, we speculated that the CWI signaling may be blocked, thereby attenuating the growth inhibition. To that end, we carried out the transcriptome sequencing experiment on the developing inflorescence stems to analyze gene expression patterns in WT, *fra1*, and *sofa1* plants. Total genes from the principal component analysis (PCA) are clearly clustered into three groups (Fig. 5A).

Among all differentially expressed genes (DEGs) in the comparisons of *fra1* versus WT (461 genes) and *sofa1* versus *fra1*

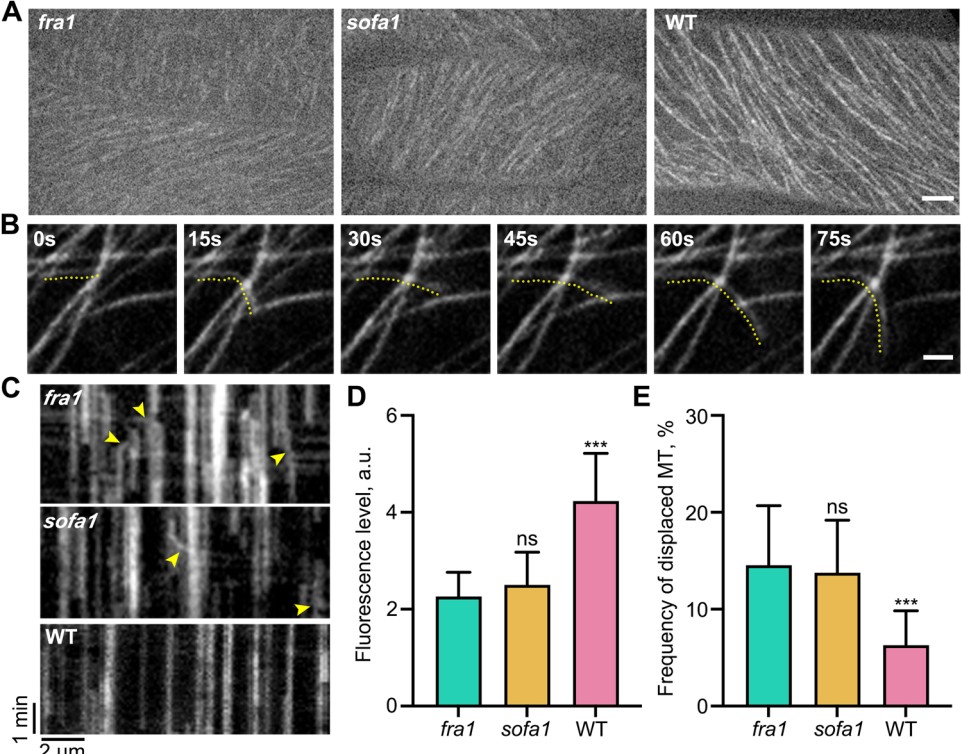

**Figure 3.  The TUB2^E69K fails to restore the lateral displacement of microtubules in *fra1*.**

(**A**) Fluorescence micrographs of mCherry-CMU1 signal in the leaf epidermal cells of *fra1*, *sofa1*, and WT plants. Scale bar indicates 5 µm. (**B**) Example of the lateral displacement of cortical microtubules labeled by VisGreen-TUB6 in the *sofa1* mutant. Scale bar indicates 2 µm. (**C**) Kymographs showing lateral displacement of cortical microtubules labeled by VisGreen-TUB6 in *fra1*, *sofa1*, and WT plants. The laterally drifting microtubules appear as slanted lines (yellow arrowheads). (**D**) Quantification of mCherry-CMU1 fluorescent intensity in the leaf epidermal cells of *fra1*, *sofa1* and WT plants. More than 100 microtubules from at least three individual cells were counted. a.u., arbitrary units. (**E**) Quantification of the frequency of displaced microtubules determined from kymographs as shown in (**C**). More than 200 microtubules from at least twelve individual cells were counted. Data information: In (**D**, **E**), data are presented as mean ± SD. ***$P < 0.001$ and ns ($P > 0.05$), no significant difference with *fra1* by one-way ANOVA. In (**D**), $P = 0.0504$ (*sofa1*), $P < 0.0001$ (WT). In (**E**), $P = 0.9259$ (*sofa1*), $P = 0.0008$ (WT). Source data are available online for this figure.

(373 genes), we found a total of 166 common genes (Fig. 5B). Intriguingly, these 166 common DEGs could be divided into two subclusters. The subcluster 1 contains 77 genes that are all down-regulated in *fra1* but maintain at WT level in *sofa1*, while the subcluster 2 contains 89 genes that are all up-regulated in *fra1* but also maintain at WT level in *sofa1* (Fig. 5C). The Gene Ontology (GO) analysis for the overlapped DEGs revealed that the most significantly enriched terms are associated with responses to various biotic and abiotic stresses, as well as some cell wall-related terms (Fig. 5D). Interestingly, most stress-related genes belong to subcluster 2 and are up-regulated in the *fra1* mutant (Fig. EV4), which commonly represents the activation of the CWI signaling pathway (Hematy et al, 2007; Zhai et al, 2024). In contrast, almost all cell wall-related genes belong to subcluster 1 and are down-regulated in the *fra1* mutant, including some xyloglucan endotransglycosylase/hydrolase (XTH) genes and an *EXPA14* (Fig. 5E). It was well established that the induced expressions of XTH and expansins (EXPA) are key players in cellular expansion (Cosgrove, 2016; Cosgrove, 2024) and the down-regulation of these cell wall genes may be associated with the dwarf phenotype in *fra1*. Based on these results, we proposed that the dwarf phenotype in *fra1* may be caused by the activation of cell wall

integrity signaling, thus restricting the growth following wall disruptions. The E69K mutation in TUB2 may block the CWI signaling, thereby alleviating growth inhibition and restoring cell elongation in *sofa1*.

In summary, we identified that the 69th residue of TUB2 plays a key role in controlling the plant height of the *fra1* mutant. When the E69K substitution occurs in TUB2 within the *fra1* mutant, the cortical microtubules show a lower growth rate and increased sensitivity to oryzalin. We proposed that the charge change of the 69th amino acid in TUB2 is likely to affect the interaction with $Mg^{2+}$, thereby influencing microtubule polymerization. However, we cannot exclude the possibility that the tubulin post-translational modifications (PTMs) play a role in microtubule dynamics. Although most PTMs occur within the carboxy-terminal tails of tubulin (Magiera and Janke, 2014), some PTMs have been detected on the amino acids at the luminal surface of microtubules, including the well-studied acetylation of lysine 40 (K40), lysine 394 (K394) in α-tubulin (Sadoul and Khochbin, 2016; Saunders et al, 2022), and lysine 252 (K252) in β-tubulin (Chu et al, 2011). The acetylation of α-tubulin K394 and β-tubulin K252 is thought to regulate microtubule stability and polymerization (Chu et al, 2011; Saunders et al, 2022). Therefore, the mutated K69 of TUB2

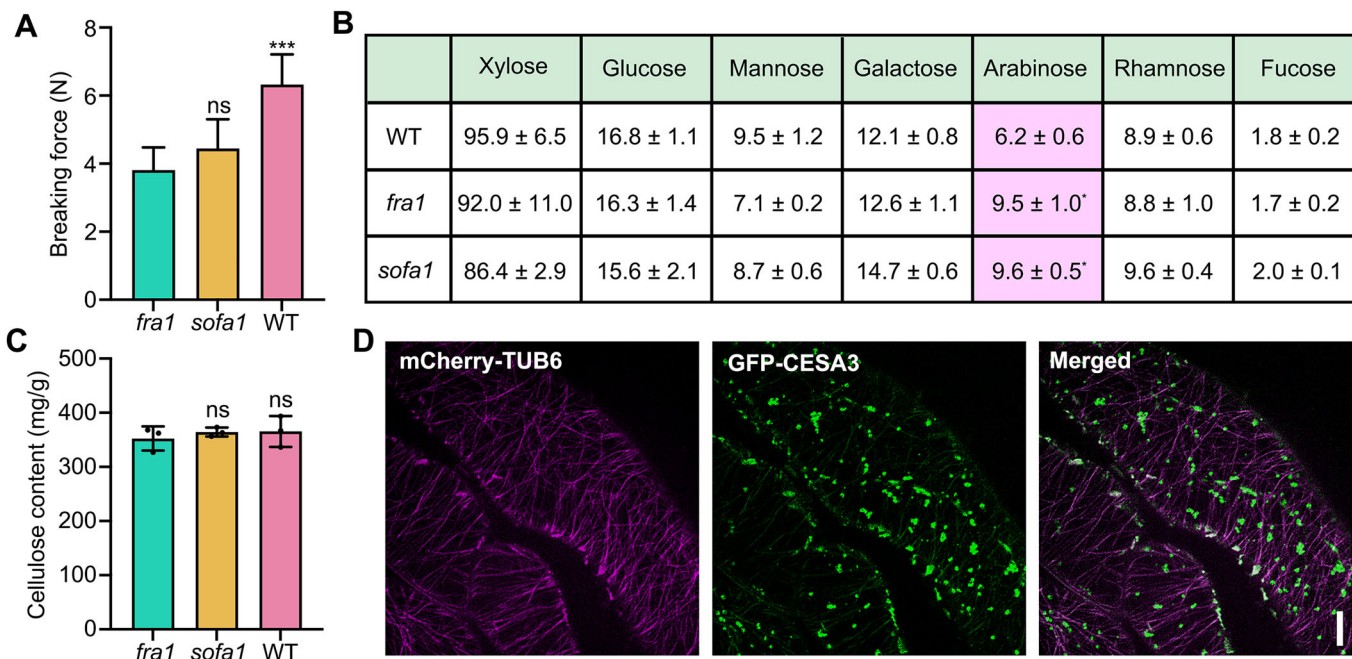

**Figure 4. Sofa1 mutants still maintain the cell wall defects as in fra1.**

(A) Quantification of breaking force of inflorescence stems in *fra1*, *sofa1* and WT plants. *n* = 10 biological replicates. (B) Cell wall monosaccharide composition of inflorescence stems in *fra1*, *sofa1* and WT plants (μg/mg). *n* = 3 biological replicates. (C) Quantification of cellulose content of inflorescence stems in *fra1*, *sofa1*, and WT plants. *n* = 3 biological replicates. (D) Dual-labeled image of mCherry-TUB6 (magenta) and GFP-CESA3 (green) in the leaf epidermal cells of *sofa1* plants. Scale bar indicates 5 μm. Data information: In (A), data are presented as mean ± SD. ***P < 0.001 and ns (P > 0.05), no significant difference with *fra1* by one-way ANOVA. P = 0.2251 (*sofa1*), P < 0.0001 (WT). In (B), data are presented as mean ± SEM. *P < 0.05 compared with WT by one-way ANOVA. P = 0.0478 (*fra1*), P = 0.0446 (*sofa1*). In (C), data are presented as mean ± SD. ns (P > 0.05), no significant difference with *fra1* by one-way ANOVA, P = 0.7862 (*sofa1*), P = 0.7469 (WT). Source data are available online for this figure.

may undergo acetylation modification, potentially altering cortical microtubule properties.

We revealed that the TUB2$^{E69K}$ fails to rescue the cell wall defects or the lateral displacement of microtubules in *fra1*, suggesting that its regulatory mechanism is, at least to some extent, independent of the known functions of FRA1. Based on transcriptomic analysis, we found a significant up-regulation of stress-related genes in the *fra1* mutant, implying that the cell wall damage caused by *FRA1*-deficiency may be monitored, which then induces the downstream signaling pathway. In contrast, some genes encoding cell wall-associated enzymes are down-regulated in the *fra1* mutant, such as some members of the XTH family (XTH8, XTH15, XTH16, XTH18, XTH33), and an α-expansin (EXPA14). *XTH8* gene has been reported to be involved in the regulation of salicylic acid (SA)-dependent dwarfism (Miura et al, 2010); XTH15::GUS is also expressed in the inflorescence stem (Becnel et al, 2006) and *xth15* mutants have reduced petiole growth rates under shade conditions (Sasidharan et al, 2010); Recently, Balkova et al have demonstrated the localization of EXPA14 in the cell wall of the shoot epidermis, suggesting its role in modulating shoot structure and function (Balkova et al, 2025). In addition, we identified a pectin methylesterase (PME41) and three fasciclin-like arabinogalactan proteins (FLA2, FLA9, FLA10). These genes may be coordinately involved in the regulation of cell elongation in the *FRA1*-deficient background.

We propose that the E69K mutation in TUB2 may block cell wall integrity signaling, which provides a possible explanation for the inhibited cell elongation in the *fra1* mutant and the restored cell elongation in the *sofa1* mutant. Our results imply the potential roles of microtubules in the cell wall integrity signaling pathway, consistent with the notion that the microtubule response to mechanical stress acts as an independent pathway, which is required to maintain the mechanical integrity of plant cells (Malivert et al, 2021). However, this model remains to be fully validated and requires further supporting evidence. It is still an exciting challenge to establish an integrated microtubule-mediated CWI signaling network, which requires elucidating how microtubules sense the wall alterations and transduce the downstream signals to make adaptive changes, and how the 69$^{th}$ residue in TUB2 contributes to monitoring the integrity of the cell wall. Alternatively, given that the TUB2$^{E69K}$ did not affect WT plants but alleviated the growth inhibition of the *fra1* mutant, it is plausible that the *fra1* mutant provides a sensitized genetic background for exploring the microtubule abnormalities. This speculation is in line with the previous finding that the dwarf phenotype of *fra1* is greatly enhanced by the knockout of *CMUs*, but knockout of *CMUs* in WT background results in no significant growth defects (Ganguly et al, 2020). Therefore, the perturbation of microtubules in the *fra1* mutant may have a more pronounced impact on the overall growth of plant cells, which could manifest as changes in plant height.

Future work can focus on these speculations to elucidate how FRA1 coordinates microtubules to control the growth of plant cells.

# Methods

## Plant materials and growth conditions

*Arabidopsis thaliana* Columbia-0 (Col-0) ecotype was used as a wild-type background in this study. A T-DNA insertion mutant, *fra1* (SALK_084463), was previously described (Kong et al, 2015). Arabidopsis plants were grown in a greenhouse at 22 °C under

**Reagents and tools table**

| Reagent/resource | Reference or source | Identifier or catalog number |
|---|---|---|
| **Experimental models** | | |
| *Agrobacterium tumefaciens*/GV3101 | This study | N/A |
| *Arabidopsis thaliana*/Col-0 | This study | N/A |
| *Arabidopsis thaliana*/fra1 | Kong et al, 2015 | SALK_084463 |
| *Arabidopsis thaliana*/sofa1 | This study | N/A |
| *Arabidopsis thaliana*/tub2$^{E69K}$ | This study | N/A |
| *Arabidopsis thaliana*/fra1 tub2c | This study | N/A |
| *Arabidopsis thaliana*/sofa1 tub2c | This study | N/A |
| **Recombinant DNA** | | |
| pTUB2-TUB2 | This study | N/A |
| pTUB2-TUB2$^{E69K}$ | This study | N/A |
| pTUB2-TUB2$^{E69R}$ | This study | N/A |
| pTUB2-TUB2$^{E69H}$ | This study | N/A |
| pTUB2-VisGreen-TUB2 | This study | N/A |
| pTUB2-VisGreen-TUB2$^{E69K}$ | This study | N/A |
| pTUB6-VisGreen-TUB6 | Liu et al, 2019 | N/A |
| pTUB6-VisGreen-TUB6$^{E69K}$ | This study | N/A |
| pCMU1-mCherry-CMU1 | This study | N/A |
| pTUB6-mCherry-TUB6 | Liu et al, 2019 | N/A |
| pCESA3-GFP-CESA3 | Kong et al, 2015 | N/A |
| pAtU6-26-sgRNA-SK | Yan et al, 2015 | N/A |
| pCAMBIA1300-pAtUBQ10:Cas9 | Yan et al, 2015 | N/A |
| **Oligonucleotides and other sequence-based reagents** | | |
| TUB2-F | CACCACATGCCACTACCATGTGTT | N/A |
| TUB2-R | ATATGGAGCTGCACTTGCCC | N/A |
| TUB2$^{E69K}$-F | TGGATTTGAAGCCTGGGACTATGGATAGTCTCA | N/A |
| TUB2$^{E69K}$-R | CCCAGGCTTCAAATCCATGAGCACTGCACGAG | N/A |
| TUB2$^{E69R}$-F | TGGATTTGCGGCCTGGGACTATGGATAGTCTCA | N/A |
| TUB2$^{E69R}$-R | CCCAGGCCGCAAATCCATGAGCACTGCACGAG | N/A |
| TUB2$^{E69H}$-F | TGGATTTGCACCCTGGGACTATGGATAGTCTCA | N/A |
| TUB2$^{E69H}$-R | CCCAGGGTGCAAATCCATGAGCACTGCACGAG | N/A |
| TUB6$^{E69K}$-F | TGGATCTTAAGCCTGGTACTATGGACAGTGTCA | N/A |
| TUB6$^{E69K}$-R | ACCAGGCTTAAGATCCATGAGAATTGCACGGG | N/A |
| TUB2-CRISPR-F | ATTGCACATCCAGGGTGGTCAATG | N/A |
| TUB2-CRISPR-R | AAACCATTGACCACCCTGGATGTG | N/A |
| pTUB2-F | CACCACATGCCACTACCATGTGTT | N/A |
| pTUB2-R | CTTGCTCACCATCTTCGGTTGGATGAGTGAAC | N/A |
| VisGreen-F | CCAACCGAAGATGGTGAGCAAGGGCGAGGA | N/A |
| VisGreen-R | TGCTCCTGCTCCCTTGTACAGCTCGTCCATGC | N/A |

| Reagent/resource | Reference or source | Identifier or catalog number |
|---|---|---|
| TUB2-VisG-F | GGAGCAGGAGCAATGCGTGAGATTCTTCACATC | N/A |
| TUB2-VisG-R | ATATGGAGCTGCACTTGCCC | N/A |
| pCMU1-F | TGACCATGATTACGAATTCCAATCTTACGGATGGACTTC | N/A |
| pCMU1-R | GCTCACCATGAATGTGTCTCTCTGTGGGAAG | N/A |
| mCherry-F | AGACACATTCATGGTGAGCAAGGGCGAGGAG | N/A |
| mCherry-R | GCTGGCATTGCGCCTGCGCCCTTGTACAGCTCGTCCATGC | N/A |
| CMU1-F | GGCGCAGGCGCAATGCCAGCAATGCCAGGTCTC | N/A |
| CMU1-R | CCAAGCTTGCATGCCTGCAGTCAGAACTTGAAACCGAGGC | N/A |
| **Chemicals, enzymes, and other reagents** | | |
| Oryzalin | Dr. Ehrenstorfer | Catalog number:DRE-C15750000 |
| Mut Express II Fast Mutagenesis Kit V2 | Vazyme | Catalog number: C214-02 |
| Phusion High-Fidelity DNA Polymerase | Thermo Fisher | Catalog number: F-530XL |
| **Software** | | |
| ImageJ | https://imagej.nih.gov/ | |
| Graphpad Prism 8 | https://www.graphpad.com/ | |
| Adobe photoshop cs5 | https://www.adobe.com | |
| **Other** | | |
| Leica M205 FA | Leica | |
| Spinning-disc confocal microscope | Perkin Elmer | |

16 h-light/8 h-dark photoperiod condition. For growth on plates, seeds were sterilized with 10% (v/v) sodium hypochlorite for 8 min and rinsed with sterile water five times, and planted on half-strength Murashige and Skoog (½ MS) medium agar plates.

## Genetic screen and MutMap-based cloning

Seeds of *fra1* mutants were incubated with 0.4% ethyl methylsulfonate (EMS) in $H_2O$ for 8 h, then rinsed in $H_2O$ 20 times, and planted in soil as M1 plants. M2 seeds were harvested from individual M1 plants and the screen was performed by scoring the suppressed dwarf phenotype of *fra1* mutant plants. The *sofa1* plant showed strong suppressor phenotype and was used in this study. The *sofa1* mutant was backcrossed to *fra1* to analyze the dominant or recessive effect of the mutation and generate the F2 progeny used for bulk sequencing. Genomic DNA from *fra*1 mutant, *sofa1* mutant, a pool of 20 F2 plants with *fra1*-like-phenotypes, a pool of 20 F2 plants with *sofa1*-like-phenotypes was used for whole-genome re-sequencing by the BSA method (Takagi et al, 2015). Genome re-sequencing was performed commercially by Novogene Technology Limited-liability Company (Beijing, China).

## Phenotype analyses

Arabidopsis plant height was measured at seven weeks after planting in soil. Measurements of leaves and siliques length were conducted by scanning to generate a digital image and then calculating by ImageJ software. Leaf samples (3rd to 5th true leaf) were dissected away from Arabidopsis plants and imaged to count the trichome branches.

To measure pith cell length, the 7-week-old basal stems were collected from *fra1*, *sofa1*, and WT plants. The stems were fixed in FAA buffer (ethyl alcohol: acetic acid: 37% formaldehyde: $H_2O$ = 10:1:2:7 volume) and then vacuumed at 4 °C. After dehydration using a gradient concentration of ethanol, the stems were embedded in historesin/hardener mix for semithin sectioning. The sections were stained with toluidine blue solution and imaged using a fully automated Stereo Microscope (Leica M205 FA) with Leica Application Suite V4.2. The pith cell length was measured using ImageJ software.

To measure the breaking force, the middle main inflorescence stems of 7-week-old plants were cut into equal-length segments. The segments were measured for their breaking forces with a tensile strength testing machine (DZ-101). The largest force required to break apart the stems was considered as the breaking force.

## Plasmid construction and plant transformation

For genetic complementation assays, the 4411 bp genomic sequence that contains a 2126 bp promoter, the *At5g62690* gene and a 343 bp 3′-untranslated region was amplified using the primers TUB2-F and TUB2-R. The amplified fragment was cloned into the pENTR vector by a TOPO based cloning strategy according to the manufacturer's instructions (Invitrogen). The resulting Entry clone was then inserted into the Gateway binary vector pEarleyGate303 by LR reaction to obtain the pTUB2 ::TUB2 construct. Site mutations of E69 in TUB2 were performed using the Mut Express II Fast Mutagenesis Kit V2 (Vazyme). The primers used to generate TUB2[E69K], TUB2[E69R], and TUB2[E69H] mutations can be found in the Reagents and Tools Table. The site mutation of E69K in TUB6 was

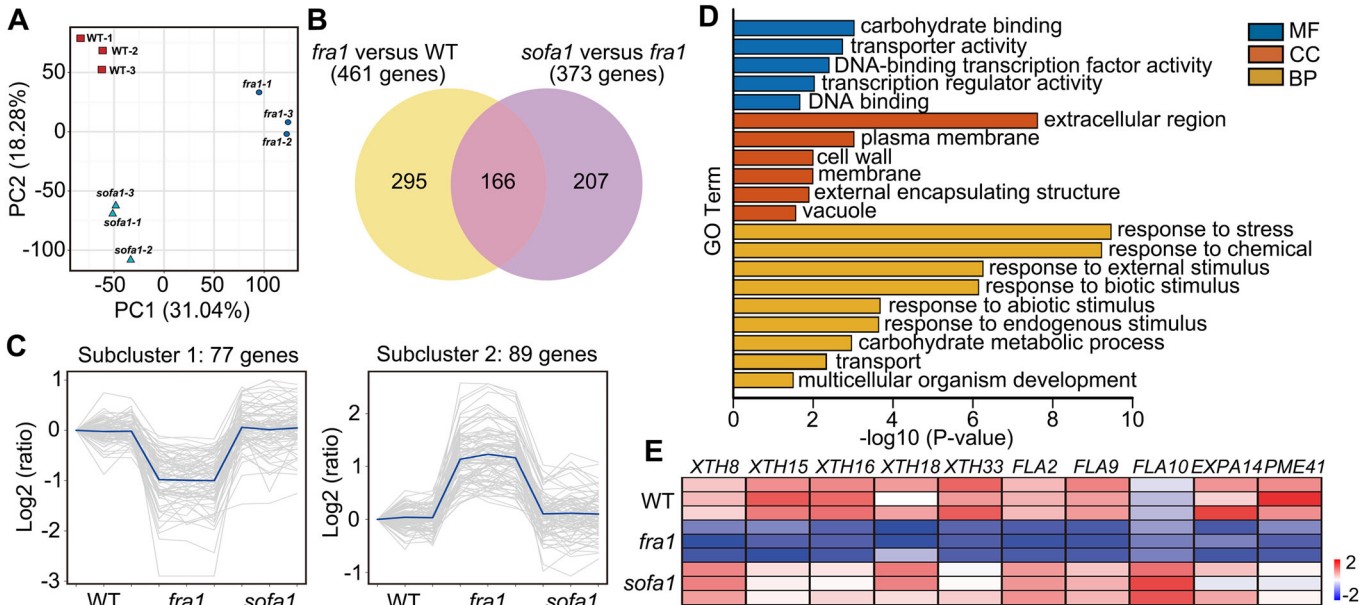

**Figure 5. Transcriptomic analysis of developing inflorescence stems in WT, fra1, and sofa1 plants.**

(A) The principal component analysis (PCA) of gene expression in WT, *fra1*, and *sofa1* inflorescence stems. Each point in the plot represents a biological replicate and each genotype includes three biological replicates. (B) The Venn diagram showing the overlapping differentially expressed genes (DEGs) in the comparisons of *fra1* versus WT and *sofa1* versus *fra1*. (C) K - means clustering analysis grouped the 166 overlapping DEGs shown in (B) into two subclusters according to their expression patterns. The number of genes in each subcluster is indicated. Each gray line represents the relative expression of DEGs and the blue lines highlight the mean expression values in each subcluster. (D) The top 20 most enriched Gene Ontology (GO) terms of the 166 overlapping DEGs. The three main GO categories of molecular functions (MF), cellular components (CC) and biological processes (BP) are shown in blue, red and yellow, respectively. (E) The heatmap showing the expression levels of cell wall-related genes in WT, *fra1*, and *sofa1* inflorescence stems. Colored bar represents z-score of log2-transformed relative expression.

generated by mutating the pTUB6::VisGreen-TUB6 marker line, which was described in our previous studies (Liu et al, 2019).

To generate pTUB2::VisGreen-TUB2, a 2126 bp promoter region of *TUB2*, the VisGreen coding sequence and the coding sequence of *TUB2* gene were amplified with the appropriate primers. The PCR fragments were linked by fusion PCR using Phusion High-Fidelity DNA Polymerase (Thermo Fisher). The resulting fusion fragment was cloned into the pENTR vector and then recombined with the Gateway binary vector pEarleyGate303. The pTUB2::VisGreen-TUB2[E69K] construct was generated by site-directed mutagenesis of the pTUB2::VisGreen-TUB2 plasmid. The pCMU1::mCherry-CMU1 construct was generated as described previously (Ganguly et al, 2020). The resulting plasmids were transformed into the *Agrobacterium* strain GV3101, followed by transformation into Arabidopsis plants by the floral dipping method.

## CRISPR/Cas9-mediated gene editing

To knock out the *TUB2* gene (*At5g62690*), we employed the CRISPR/Cas9 genome editing technology using a *YAO* promoter-driven system (Yan et al, 2015). A single guide RNA (sgRNA) site (CACATCCAGGGTGGTCAATG) was first designed. It was incorporated into PCR forward and reverse primers, respectively, to create the sgRNA scaffold for mutation. The sgRNA scaffold was inserted into *Bsa*I-linearized AtU6-26-sgRNA-SK. Then the cassette containing the sgRNA was cloned into the pCAM-BIA1300-pAtUBQ10:Cas9 binary vector. The construct was

transferred into *fra1* plants and *sofa1* plants using *Agrobacterium* strain GV3101 and medium supplemented with hygromycin (30 mg/l) was used to select transformants.

## Molecular model of TUB2 and TUB2[E69K]

The three-dimensional (3D) molecular models of TUB2 and TUB2[E69K] were constructed using SWISS-MODEL software (https://www.swissmodel.expasy.org/). A resolved crystal structure of 4FFB with 73.67% sequence identity was selected as the template, whose β-tubulin binds a $Mg^{2+}$ ligand and a GTP ligand (Ayaz et al, 2012). Visualization and comparison of the TUB2 protein and TUB2[E69K] protein were performed through PyMOL program.

## Confocal microscopy and image processing

The VisGreen and mCherry fluorescence in leaf cells was detected using a spinning-disc confocal microscope (UltraViewVoX, Perkin Elmer, Beaconsfield, UK) equipped with the Yokogawa Nipkow CSU-X1 spinning disc scanner, Hamamatsu EMCCD 9100-13, Nikon TiE inverted microscope, as described previously (Liu et al, 2014). The VisGreen and mCherry fluorescence was excited by the 488 nm and 561 nm lasers respectively. All microscopy was performed using a 60-fold immersion objective.

To quantify the fluorescence intensity of microtubules labeled by VisGreen-TUB2 or VisGreen-TUB2[E69K], all images were acquired with identical settings. And the mean fluorescent signal per unit area of cortical microtubule was measured using the ImageJ

software. To quantify the microtubule density, a 10-μm line was drawn perpendicular to the predominant orientation of cortical microtubules within the cell, and the number of microtubules across this line was quantified. To quantify microtubule dynamics, time-lapse images were captured at 5 s intervals for 4 min. The walking average plugin was used to reduce the noise in time-lapse series. The parameters of cortical microtubule plus ends were extracted manually using kymograph analysis in ImageJ software. Quantitative measurement of the fluorescence intensity of mCherry-CMU1 and the frequency of displaced microtubules were conducted as described previously (Ganguly et al, 2020).

### Oryzalin treatments

For microtubule-disrupting drugs oryzalin treatments, seeds were sterilized and sown on ½ MS medium agar plates. Then 6-day-old seedlings with an emerging pair of primary leaves were grown on ½ MS medium containing oryzalin at the indicated concentrations for 24 h. Seedlings placed on medium containing DMSO (the solvents used for the oryzalin) were used as mock. The fluorescent signal of microtubules was examined in the abaxial epidermal cells of the central vein in primary leaves. For root analyses, 6-day-old *fra1* and *sofa1* seedlings were grown vertically and then exposed for 24 h to various concentrations of oryzalin. The root length is determined by measuring the distance from the root tip to the emergence of root hairs. The root width denotes bottom width of the root mature zone.

### Cell wall composition analysis

To prepare cell wall materials, mature inflorescence stems of wild-type, *fra1* plants and *sofa1* plants were collected, with three biological replicates, each replicate containing at least fifteen plants. After grounding into powder, the samples were washed with 70% ethanol and further extracted with methanol/chloroform (1:1, v/v) and acetone. Starch was further removed using amylase and pullulanase. The resulting cell wall residues were dried in a vacuum oven and used to analyze of cell wall composition. The cellulose content was measured using the Updegraff method (Updegraff, 1969). The monosaccharide composition were analyzed by high-performance anion-exchange chromatography (HPAEC) on a CarboPac PA-20 anion-exchange column (3 by 150 mm; Dionex) using a pulsed amperometric detector (PAD; Dionex ICS 5000 system) by Sanshu Biotech. Co., LTD (Shanghai, China).

### Transcriptome sequencing and data analysis

For RNA-Seq, the middle main inflorescence stems of 7-week-old wild-type, *fra1* plants and *sofa1* plants were collected, and total RNA was extracted using the RNAprep Pure Plant Kit (Tiangen, Beijing, China) according to the instructions provided by the manufacturer. Three biological replicates were performed, with each replicate having more than 20 individual stems. The RNA quantification, qualification, library preparation, and data analysis were performed by Biomarker Technologies Co., Ltd (Beijing, China). Differential expression analysis was performed using DESeq2, and genes with an adjusted *P* value < 0.01 and Fold Change ≥2 found by DESeq2 were assigned as differentially expressed. Gene Ontology enrichment analysis of the differentially

expressed genes was implemented by the clusterProfiler packages based on Wallenius non-central hyper-geometric distribution (Young et al, 2010).

### Quantification and statistical analysis

The statistical analyses were performed using GraphPad Prism 8 software. Statistical significance in pair-wise comparisons was evaluated by Student's *t* test (*t* test). One-way ANOVA and Dunnett's test were used to calculate the statistical significance of multiple groups of samples. Asterisks indicate statistical significance: ***$P < 0.001$, **$P < 0.01$, *$P < 0.05$. All experiments were repeated at least three times, and representative results are shown. Specific descriptions of statistical tests are provided in the figure legends.

## Data availability

The RNA-seq data produced in this study is available: Sequence Read Archive PRJNA1182924.

The source data of this paper are collected in the following database record: biostudies:S-SCDT-10_1038-S44319-025-00507-4.

## Peer review information

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

## Acknowledgements

The authors appreciate Dr. Yihua Zhou and Dr. Baocai Zhang (Institute of Genetics and Developmental Biology, Chinese Academy of Sciences) for the measurement of stems breaking force. The authors are grateful to Ms. Yao Wu, Dr. Lei Su and Ms. Haiyun Wang (Institute of Microbiology, Chinese Academy of Sciences) for providing technical assistance and training service. Thanks Prof. YI Zhang for helpful discussion. This study was supported by the

National Science Fund for Distinguished Young Scholars (Grant No. 31925003) and the National Natural Science Foundation of China (NSFC, Grant No. 31771496).

## Author contributions

**Huanhuan Yang**: Conceptualization; Formal analysis; Validation; Investigation; Visualization; Writing—original draft. **Jie Wang**: Investigation. **Guangda Wang**: Formal analysis; Visualization; Writing—review and editing. **Chaofeng Wang**: Resources. **Xiaxia Zhang**: Project administration. **Juan Tian**: Project administration. **Yanjun Yu**: Project administration. **Zhaosheng Kong**: Conceptualization; Resources; Supervision; Funding acquisition; Writing—review and editing.

Source data underlying figure panels in this paper may have individual authorship assigned. Where available, figure panel/source data authorship is listed in the following database record: biostudies:S-SCDT-10_1038-S44319-025-00507-4.

## Disclosure and competing interests statement

The authors declare no competing interests.

# Expanded View Figures

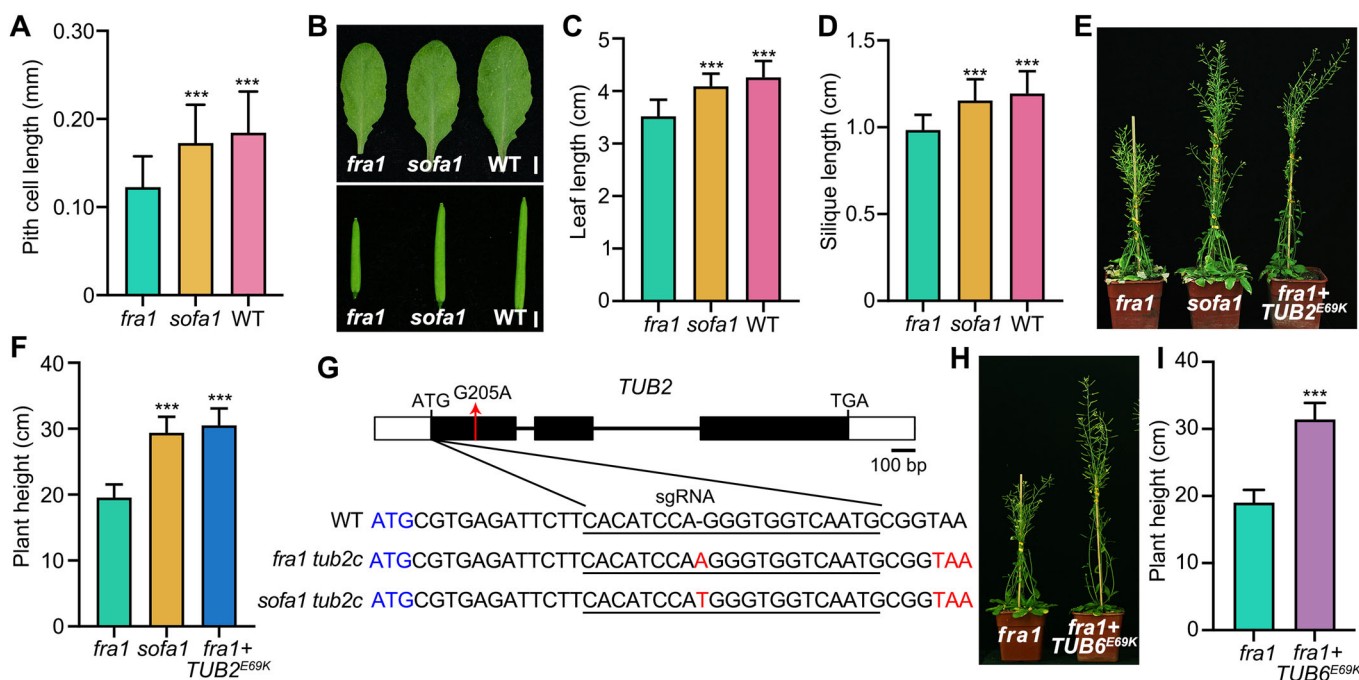

**Figure EV1. Identification and gene mapping of the suppressor of *fra1*.**

(A) Quantification of the pith cell length as shown in Fig. 1C. 50 cells from five individual plants were counted. (B) The leaves and siliques of *fra1*, *sofa1* and WT plants. Scale bars indicate 5 cm (upper) and 2 cm (lower), respectively. (C, D) Quantification of the length of leaves (C) and siliques (D) of *fra1*, *sofa1* and WT plants as shown in (B). 20 leaves and 50 siliques from five individual plants were counted. (E, F) Growth phenotype of *fra1*, *sofa1* and *fra1* + *TUB2*^E69K plants (E) and quantification of their plant height (F). More than 20 plants were counted for each sample. (G) A schema of the *TUB2* gene showing the CRISPR/Cas9 targeted site in different mutant lines. The single guide RNA (sgRNA) site is indicated by underlines. The start codons (ATG) are shown in blue letters; base insertion (A/T) and premature stop codons (TAA) are denoted with red letters. (H, I) Growth phenotype of *fra1*, *fra1* + *TUB6*^E69K plants (H) and quantification of their plant height (I). More than 20 plants were counted for each sample. Data information: In (A, C, D, F), data are presented as mean ± SD. \*\*\*$P < 0.001$ compared with *fra1* by one-way ANOVA. In (A), $P < 0.0001$ (*sofa1*), $P < 0.0001$ (WT). In (C), $P = 0.0005$ (*sofa1*), $P < 0.0001$ (WT). In (D), $P < 0.0001$ (*sofa1*), $P < 0.0001$ (WT). In (F), $P < 0.0001$ (*sofa1*), $P < 0.0001$ (*fra1* + *TUB2*^E69K). In (I), data are presented as mean ± SD. \*\*\*$P < 0.001$ compared with *fra1* by Student's *t* test, $P < 0.0001$.

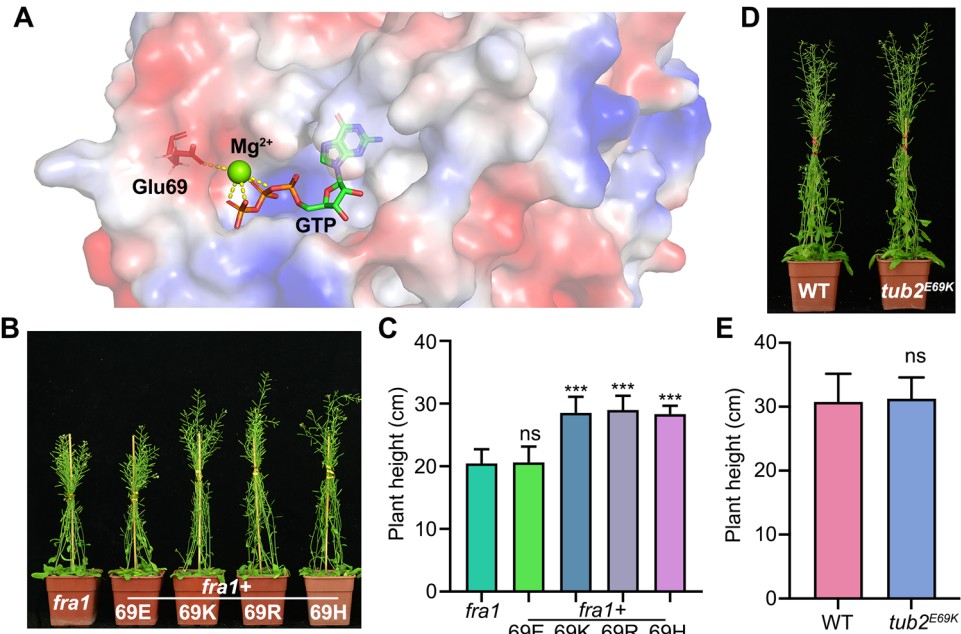

**Figure EV2. The 69th residue of TUB2 plays a key role in plant height control of *fra1*.**

(A) The resolved crystal structure of yeast β-tubulin (PDB:4FFB), accommodating a Mg²⁺ ion (green sphere) and a GTP (sticks colored by atom type). The 69Glu (red stick) is highlighted. (B, C) Growth phenotype of *fra1*, *fra1* + TUB2^69E/K/R/H (*fra1* + 69E, 69K, 69 R, 69H) plants (B) and quantification of their plant height (C). More than 20 plants were counted for each sample. (D, E) Growth phenotype of WT and *tub2^E69K* plants (D) and quantification of their plant height (E). More than 20 plants were counted for each sample. Data information: In (C), data are presented as mean ± SD. ***P < 0.001 and ns (P > 0.05), no significant difference with *fra1* by one-way ANOVA, P = 0.9974 (69E), P < 0.0001 (69K), P < 0.0001 (69 R), P < 0.0001 (69H). In (E), data are presented as mean ± SD. ns (P > 0.05), no significant difference with WT by Student's *t* test, P = 0.1026.

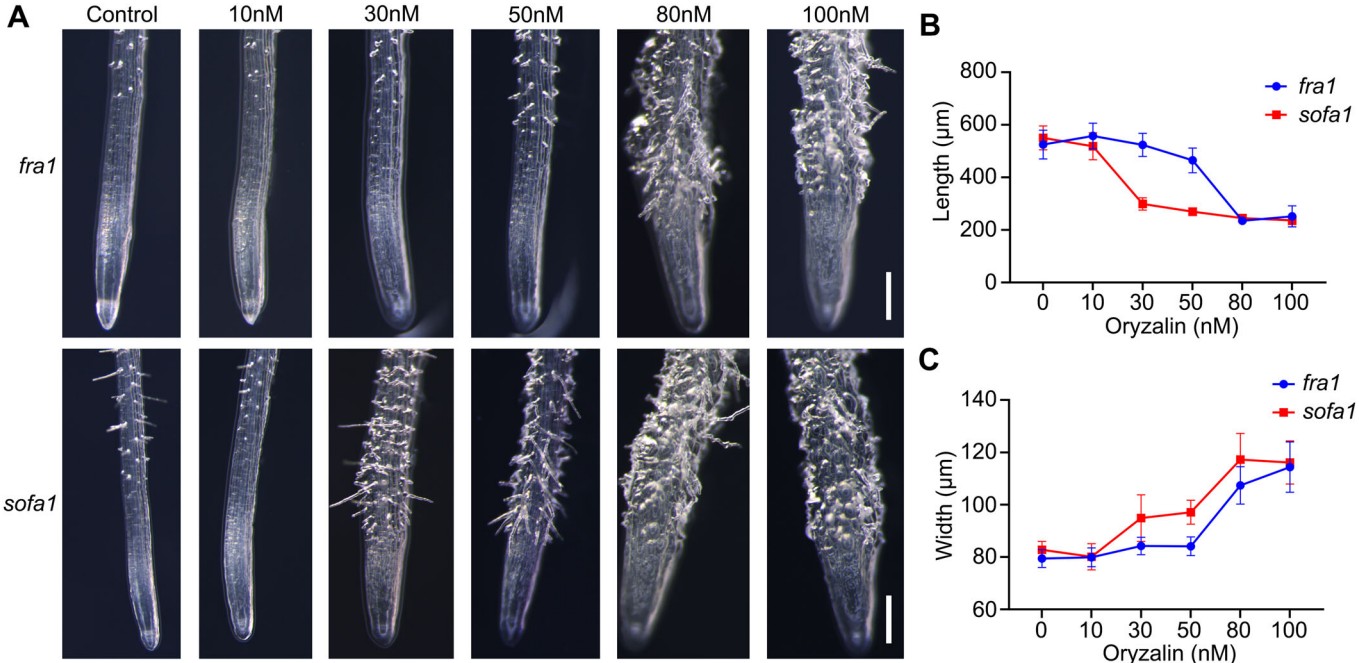

**Figure EV3. The incorporation of TUB2$^{E69K}$ influences the stability of microtubules.**

(A) Primary roots of 6-day-old *fra1* and *sofa1* seedlings exposed for 24 h to various concentrations of oryzalin. Scale bars indicate 100 μm. (B, C) Quantification of the root length (B) and root width (C) as shown in (A). Data are presented as mean ± SD. *n* = 8 biological replicates.

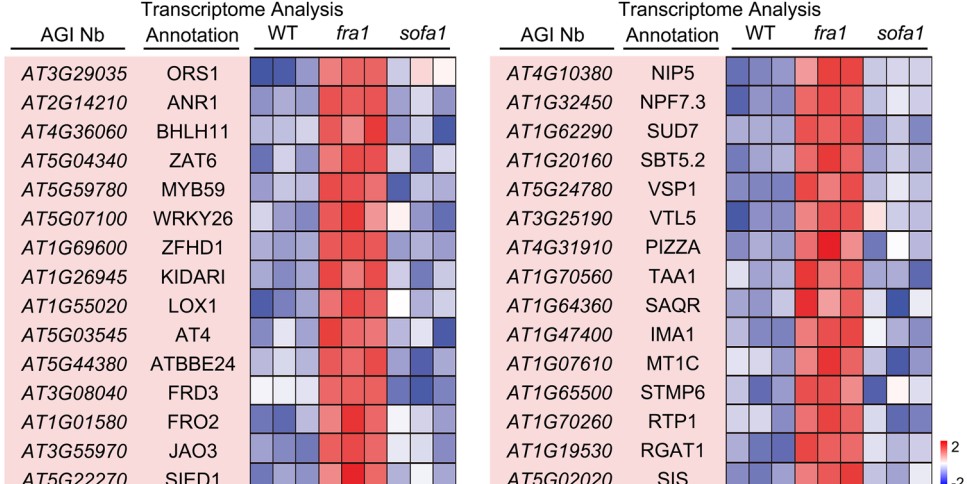

**Figure EV4. The expression levels of stress-related genes in WT, *fra1* and *sofa1*.**

The heatmap showing the expression levels of a total of 30 stress-related genes in WT, *fra1* and *sofa1* inflorescence stems. Colored bar represents z-score of log2-transformed relative expression.

