## [Peer Review File · EMBO Reports]

E69K mutation in β -Tubulin 2 blocks cell wall integrity signaling during plant cell elongation

Huanhuan Yang, Jie Wang, Guangda Wang, Chaofeng Wang, Xiaxia Zhang, Juan Tian, Yanjun Yu, and Zhaosheng Kong

Corresponding author(s): Zhaosheng Kong (zskong@im.ac.cn)

Review Timeline:

Submission Date:	3rd Dec 24
Editorial Decision:	5th Feb 25
Revision Received:	21st Apr 25
Editorial Decision:	4th Jun 25
Revision Received:	5th Jun 25
Accepted:	6th Jun 25

Editor: Deniz Senyilmaz Tiebe

Transaction Report:

Dear Prof. Kong,

Thank you for transferring your manuscript to EMBO Reports. My apologies for this unusual delay in getting back to you. Three referees agreed to review your manuscript. So far, we have received two referee reports that are copied below. Given that both referees are in fair agreement that you should be given a chance to revise the manuscript, I would like to ask you to begin revising your study along the lines suggested by the referees.

Please note that this is a preliminary decision made in the interest of time, and that it is subject to change should the third referee offer very strong and convincing reasons for this. As soon as we receive the final report on your manuscript, we will forward it to you as well.

Referees express interest in the proposed effect of β -TubulinE69K mutation on cell wall signaling. However, they also raise some concerns that need to be addressed to consider publication here. In particular,

- More (epistatic) evidence is required to lend further support to the proposed effect of β -TubulinE69K mutation on cell wall signaling (referee #2, major point 1).
- The effects of sofa1 and fra1 mutations on cell division rates should be explored (referee #1, major point 1)
- The effect of sofa1 mutation on tubulin turnover rates needs to be investigated (referee #1, major point 2)
- Alternative explanations as to how sofa1 mutation can affect microtubule stability should be discussed (referee #1, major point 3).

Given these recommendations, we would like to invite you to revise your manuscript with the understanding that the referee concerns (as in their reports) must be fully addressed and their suggestions taken on board. Please address all referee concerns in a complete point-by-point response. Acceptance of the manuscript will depend on a positive outcome of a second round of review. It is EMBO reports policy to allow a single round of major experimental revision only and acceptance or rejection of the manuscript will therefore depend on the completeness of your responses included in the next, final version of the manuscript.

We realize that it is difficult to revise to a specific deadline. In the interest of protecting the conceptual advance provided by the work, we recommend a revision within 3 months. Please discuss the revision progress ahead of this time with me if you require more time to complete the revisions, or if you have questions or comments regarding the revision (also by video chat).

1. A data availability section providing access to data deposited in public databases is missing (where applicable).
2. Your manuscript contains statistics and error bars based on $n=2$. Please use scatter plots in these cases.

You can submit the revision either as a Scientific Report or as a Research Article. For Scientific Reports, the revised manuscript can contain up to 5 main figures and 5 Expanded View figures, and it should not exceed 27000 characters. If the revision leads to a manuscript with more than 5 main figures it will be published as a Research Article. In this case the Results and Discussion section should be separate. If a Scientific Report is submitted, these sections have to be combined. This will help to shorten the manuscript text by eliminating some redundancy that is inevitable when discussing the same experiments twice. In either case, all materials and methods should be included in the main manuscript file.

4) a .docx formatted letter INCLUDING the reviewers' reports and your detailed point-by-point responses to their comments. As part of the EMBO publication's Transparent Editorial Process, EMBO reports publishes online a Review Process File (RPF) to accompany accepted manuscripts. This File will be published in conjunction with your paper and will include the referee reports, your point-by-point response and all pertinent correspondence relating to the manuscript.

<https://www.embopress.org/page/journal/14693178/authorguide#transparentprocess>

5) a complete author checklist, which you can download from our author guidelines

<https://www.embopress.org/page/journal/14693178/authorguide>. Please insert information in the checklist that is also reflected in the manuscript. The completed author checklist will also be part of the RPF.

6) Please note that all corresponding authors are required to supply an ORCID ID for their name upon submission of a revised manuscript (<<https://orcid.org/>>). Please find instructions on how to link your ORCID ID to your account in our manuscript tracking system in our Author guidelines

<<https://www.embopress.org/page/journal/14693178/authorguide#authorshippinguidelines>>

7) Before submitting your revision, primary datasets produced in this study need to be deposited in an appropriate public database (see <https://www.embopress.org/page/journal/14693178/authorguide#datadeposition>). Please remember to provide a reviewer password if the datasets are not yet public. The accession numbers and database should be listed in a formal "Data Availability" section placed after Materials & Method (see also

<https://www.embopress.org/page/journal/14693178/authorguide#datadeposition>). Please note that the Data Availability Section is restricted to new primary data that are part of this study. * Note - All links should resolve to a page where the data can be accessed. *

Additional information on source data and instruction on how to label the files are available:

<https://www.embopress.org/page/journal/14693178/authorguide#sourcedata>

9) Our journal encourages inclusion of *data citations in the reference list* to directly cite datasets that were re-used and obtained from public databases. Data citations in the article text are distinct from normal bibliographical citations and should directly link to the database records from which the data can be accessed. In the main text, data citations are formatted as follows: "Data ref: Smith et al, 2001" or "Data ref: NCBI Sequence Read Archive PRJNA342805, 2017". In the Reference list, data citations must be labeled with "[DATASET]". A data reference must provide the database name, accession number/identifiers and a resolvable link to the landing page from which the data can be accessed at the end of the reference. Further instructions are available at <http://www.embopress.org/page/journal/14693178/authorguide#referencesformat>

10) Regarding data quantification (see Figure Legends:

<https://www.embopress.org/page/journal/14693178/authorguide#figureformat>)

- the name of the statistical test used to generate error bars and P values,

- the number (n) of independent experiments (please specify technical or biological replicates) underlying each data point,

- the nature of the bars and error bars (s.d., s.e.m.),

- If the data are obtained from n Program fragment delivered error ``Can't locate object method "less" via package "than" (perhaps you forgot to load "than"?) at //ejpvfs23/sites23b/embor_www/letters/embor_decision_revise_and_review.txt line 56.' 2, use scatter blots showing the individual data points.

12) Please also note our reference format:

13) All Materials and Methods need to be described in the main text using our 'Structured Methods' format, which is required for all research articles. According to this format, the Methods section includes a Reagents and Tools Table (listing key reagents, experimental models, software and relevant equipment and including their sources and relevant identifiers) followed by a Methods and Protocols section describing the methods using a step-by-step protocol format. The aim is to facilitate adoption of the methodologies across labs. More information on how to adhere to this format as well as a downloadable template (.docx) for the Reagents and Tools Table can be found in our author guidelines:

I look forward to seeing a revised version of your manuscript when it is ready. Please let me know if you have questions or comments regarding the revision.

Kind regards,

Deniz Senyilmaz Tiebe

Deniz Senyilmaz Tiebe, PhD
Senior Scientific Editor
EMBO Reports

Referee #1:

Yang et al. address the important and timely question of how plants sense cell wall integrity using the microtubule cytoskeleton and subsequently activate transcriptional responses to inhibit growth in face of mechanical stress. The authors report the single key finding that the mutant β -tubulin TUB2E69K blocks cell wall integrity signaling in Arabidopsis.

Using the previously published fra1 mutant as a tool in a genetic screen, the authors discover a suppressor line named sofa1. By whole-genome sequencing, they pinpoint the suppressor mutation to a single amino acid substitution in the Arabidopsis TUB2 gene. They present convincing evidence of the importance of this single negatively charged amino acid (Glu69) for inhibition of growth in the fra1 background. Upon a dominant gain-of-function E69K substitution, microtubules are more unstable with a slower growth rate and increased response to depolymerization by oryzalin. Interestingly, the authors show that sofa1 mutants have similar cell wall properties and lateral instability of microtubules to the fra1 mutant. Last, by RNA sequencing, the authors show that the sofa1 mutation, i.e., the E69K substitution in TUB2, likely blocks the cell wall integrity signaling that is responsible for growth inhibition in fra1 plants. In summary, the research is novel and scientifically sound, and the manuscript is interesting to the general molecular biology audience. The single key finding is also robustly supported by independent lines of experimental evidence, including data from the organism to the molecular level, a major strength of this paper. However, I have three major points and some minor points that should be addressed before publication of this manuscript.

Major points:

- 1) Is there no evidence for a change in cell division rates in fra1 and sofa1 mutants that could account for part of the phenotypes observed? If there is a change in cell division rates, what is the contribution of this change to the observed phenotypes? Microtubules are central to both cell expansion and division in plants. Thus, these questions should be clarified, and the answers are crucial to support the link between the phenotypes observed and the alterations in cell wall synthesis detected by the RNA sequencing in fra1 mutants.
- 2) The signal of TUB2E69K looks spottier in comparison to the wild-type TUB2 (Figure 3A). The E69K mutation indeed does not

completely disrupt incorporation of TUB2 into the MTs, but it could decrease its incorporation rate/ratio. I recommend that the authors look at the tubulin turnover rates of the two tubulin constructs by using FRAP.

3) I still have trouble understanding exactly how the Glu69 site in TUB2 can account for sensing of cell wall integrity and subsequent signaling. The authors suggest that the exchange from a negatively charged amino acid to a positively charged amino acid can affect the interaction of TUB2 with Mg²⁺ and therefore affect microtubule polymerization and GTP hydrolysis. There are other possibilities that need to be mentioned. For example, post-translation modifications (PTMs) - glutamate can be (poly)glutamylated, whereas lysine cannot. In fact, polyglutamylated is a common PTM of tubulin. On other hand, lysine can undergo PTMs that glutamate cannot (e.g., acetylation).

Minor points:

1) Lines 166 and 167 - I understand what the authors mean, but it is unclear from the text which hypothesis the authors are testing here. Please rewrite.

2) Lines 227 to 228 (and Figure 3F) - The oryzalin concentrations are quite high, in the micromolar range. In my experience, an application of oryzalin in the nanomolar range is enough to depolymerize microtubules. Can the authors confirm this is the correct concentration, and perhaps explain why such high concentrations were used? In the other experiments where they look at root swelling (Figure S3), the authors also use oryzalin in nanomolar concentrations.

3) Figure 3F - The authors should provide a quantification of the disappearance of microtubules upon treatment with oryzalin for 24 hours in TUB2 in comparison to TUB2E69K plants. They state 5 biological replicates were tested, but no other numbers are shown.

4) Figure 4B - I have trouble seeing the lateral displacement the authors describe. Are the yellow dots covering the displaced microtubule? If yes, this microtubule should be shown in a different way. Alternatively, a video could be uploaded to show this event.

5) Figure 4C - It looks like some lateral displacement events in the WT were not assigned with yellow arrowheads. The authors should mark these events appropriately not to mislead the reader.

6) Grammar/spelling should be checked throughout the manuscript (for example line 388 'deficiency' instead of 'defeciency')

Referee #2:

The manuscript is very interesting; the discovery of TUB2 E69K mutation and its ability to restore the growth defect of *fra1* mutant will advance the knowledge on microtubule dynamics and cell wall integrity signalling. The authors provided solid genetic evidence, but the regulatory mechanisms were not fully explored. I understand that studies published in EMBO Reports may prefer novelty and functional insight over extensive mechanistic detail, but still, I believe adding some additional evidence on cell wall integrity signalling or more insightful discussion will be helpful. Please see my specific comments below:

Major issue.

1. The idea that TUB2E69K blocks the regulation of CWI gene expression in *fra1* mutant is compelling; however, this conclusion is only supported by the RNA-seq data. If possible, the author may consider KO some key genes in the CWI signalling pathway; could this partially rescue the *fra1* mutant phenotype? Alternatively, some chemical or stress treatment will also be helpful (I don't have any ideas in mind; treat the plant with something that relies on CWI-signaling, but the *sofa* mutant is less sensitive?)

Minor issues

1. Figure 3F requires some quantification of microtubule density.

2. Figure 4. The authors should provide additional evidence on the level of CUM1 in different genetic backgrounds. e.g qPCR to prove the expression of CUM is equal at the transcriptional level, or an immune-blot? The level of CUM should be identical in all backgrounds, but it might be miss-localized or degraded in the *fra1* or *sofa1* mutant? Also, it should be mentioned in the text as CUM1-labelled microtubules (lanes 237-256).

3. CUM1 was previously known as KLCR1 (kinesin light chain-related protein, Burstenbinder J Biol Chem 2013), but its name became slightly complicated when a group gave it a different name CUM for some reason (Liu et al., Dev Cell 2016). For good practice, both names should be used (or at least mentioned) in the manuscript (see Zang et al. Curr Biol, 2021). And some original papers related to KLCR/CUM should also be discussed.

Point-by-point responses:**Referee #1:**

Yang et al. address the important and timely question of how plants sense cell wall integrity using the microtubule cytoskeleton and subsequently activate transcriptional responses to inhibit growth in face of mechanical stress. The authors report the single key finding that the mutant β -tubulin TUB2E69K blocks cell wall integrity signaling in Arabidopsis.

Using the previously published *fra1* mutant as a tool in a genetic screen, the authors discover a suppressor line named *sofa1*. By whole-genome sequencing, they pinpoint the suppressor mutation to a single amino acid substitution in the Arabidopsis TUB2 gene. They present convincing evidence of the importance of this single negatively charged amino acid (Glu69) for inhibition of growth in the *fra1* background. Upon a dominant gain-of-function E69K substitution, microtubules are more unstable with a slower growth rate and increased response to depolymerization by oryzalin. Interestingly, the authors show that *sofa1* mutants have similar cell wall properties and lateral instability of microtubules to the *fra1* mutant. Last, by RNA sequencing, the authors show that the *sofa1* mutation, i.e., the E69K substitution in TUB2, likely blocks the cell wall integrity signaling that is responsible for growth inhibition in *fra1* plants. In summary, the research is novel and scientifically sound, and the manuscript is interesting to the general molecular biology audience. The single key finding is also robustly supported by independent lines of experimental evidence, including data from the organism to the molecular level, a major strength of this paper. However, I have three major points and some minor points that should be addressed before publication of this manuscript.

Response: Thank you very much for your positive evaluation on our work. As suggested, we have carefully revised the manuscript and addressed all the comments and concerns.

Major points:

1) Is there no evidence for a change in cell division rates in *fra1* and *sofa1* mutants that could account for part of the phenotypes observed? If there is a change in cell division rates, what is the contribution of this change to the observed phenotypes? Microtubules are central to both cell expansion and division in plants. Thus, these questions should be clarified, and the answers are crucial to support the link between the phenotypes observed and the alterations in cell wall synthesis detected by the RNA sequencing in *fra1* mutants.

Response: Thank you for your insightful comments. As you correctly pointed out, cell division and cell expansion coordinately determine organ growth, which both rely on the function of microtubules. Based on the observation that the pith cells in *sofa1* mutants are longer than those in *fra1* mutants (Figures 1C and EV1A), we propose that the restored plant height in the *sofa1* plants is caused by the restoration of cell expansion (Lines 109-110). Notably, there is a

strong positive correlation between stem length and cell length in *fra1* and *sofa1* mutants (Please see Responsive Figure 1A), highlighting the pivotal role of cell elongation in determining plant height.

To further investigate the potential contribution of cell division, we performed transverse sections of the stems of WT, *fra1* and *sofa1* plants and counted the number of cambium cells. No significant differences were observed among the three genotypes (Please see Responsive Figure 1B-D). Furthermore, we analyzed our transcriptome data of developing inflorescence stems and found that most cell division-related genes were not differentially expressed (Please see Responsive Figure 1E). Additionally, we counted the number of root meristematic cells in seedlings and again found no differences among these three genotypes (Please see Responsive Figure 1F and 1G). Based on these results, we conclude that the changes in plant height in *fra1* and *sofa1* mutants are primarily due to alterations in cell elongation, with minimal or undetectable contribution from changes in cell division.

Responsive Figure 1. Analysis of the contribution of cell division to the phenotypes in *fra1* and *sofa1* mutants.

(A) The relative stem length and cell length of WT and *sofa1* plants, with *fra1* set as the reference (relative length = 1). The data are based on the quantification of plant height (Figure 1B) and quantification of the pith cell length (Figure EV1A) across the three genotypes. Data are presented as mean \pm SD.

(B) Transverse sections of developing inflorescence stems in WT, *fra1*, and *sofa1*

stained with toluidine blue. Scale bars indicate 100 μm .

(C) Enlarged view of vascular tissue in stem sections, labeling phloem, fascicular cambium, and xylem.

(D) Quantification of cambium cell number in WT, *fra1*, and *sofa1*. Error bars indicate SD (n=15). ns ($p > 0.05$), no significant difference with WT by One-way ANOVA.

(E) Heatmap showing expression levels of cell cycle-related genes in WT, *fra1*, and *sofa1* inflorescence stems.

(F) FM4-64 staining of roots in 6-day-old WT, *fra1*, and *sofa1* seedlings. White arrows indicate the quiescent center (QC) and root meristem boundary. Scale bar indicates 20 μm .

(G) Quantifications of the meristematic cell number of three genotypes seedlings as shown in (F). Error bars indicate SD (n=16). ns ($p > 0.05$), no significant difference with WT by One-way ANOVA.

2) The signal of TUB2^{E69K} looks spottier in comparison to the wild-type TUB2 (Figure 3A). The E69K mutation indeed does not completely disrupt incorporation of TUB2 into the MTs, but it could decrease its incorporation rate/ratio. I recommend that the authors look at the tubulin turnover rates of the two tubulin constructs by using FRAP.

Response: Thank you for your careful review of our work and professional suggestions. We indeed noticed that the TUB2^{E69K} signal appeared spottier compared to the wild-type TUB2. To quantify this difference, we measured the fluorescence intensity of microtubules labeled by VisGreen-TUB2 or VisGreen-TUB2^{E69K} in *fra1*, and found that TUB2^{E69K} signals were reduced to ~50% of TUB2 levels (Please see Responsive Figure 2A). This result indicates that the TUB2^{E69K} could incorporate into microtubules by polymerization at plus-ends, but in smaller amounts than wild-type TUB2.

Following your recommendation, we conducted FRAP experiments on pavement cells to compare the turnover rates of the two tubulin constructs. After photobleaching, the fluorescence recovery occurred uniformly throughout the bleached region (Please see Responsive Figure 2D), indicating the highly dynamic nature of cortical microtubules. However, we found that there was no significant difference in the recovery halftime ($T_{1/2}$) after photobleaching between TUB2 (24.11s) and TUB2^{E69K} (26.02s) (Please see Responsive Figure 2B and 2C). The FRAP technique is known to assess the behavior of a population of microtubules (Hush *et al*, 1994), and the resulting turnover rates reflect the **overall dynamic instability** of microtubules, including growth, shrinkage, and the frequency of transitions between these two states (Shaw *et al*, 2003; Vorobjev *et al*, 1999). Therefore, the similar tubulin turnover rates ($T_{1/2}$) indicate that the E69K mutation in TUB2 fails to affect the overall level of microtubule dynamic activity, and the spottier signal of TUB2^{E69K} is not caused by the reduced turnover rates compared to TUB2. This finding is consistent with our results that TUB2^{E69K} does not disrupt overall organization of cortical

microtubules (Figure 2A).

We think the FRAP technique may serve as an assessment of dynamic stability of microtubules, and the fluorescence intensity may better reflect the incorporation of TUB2^{E69K} into microtubules. Therefore, we have added the fluorescence intensity to **Figure 2B** and provided corresponding descriptions in the revised manuscript. Please see lines 184-187.

Responsive Figure 2. Reduced incorporation of TUB2^{E69K} into cortical microtubules than wild-type TUB2.

(A) Fluorescence intensity of cortical microtubules labeled by VisG-TUB2 and VisG-TUB2^{E69K} in *fra1* mutants. Error bars indicate SD. ***p < 0.001 compared with *fra1*+TUB2 by Student's t test. n=16 cells from eight individual plants. a.u., arbitrary units.

(B, C) Intensity profile of fluorescence recovery after photobleaching (FRAP) of VisG-TUB2 (B) and VisG-TUB2^{E69K} (C) in pavement cells. Five images were taken before the photobleaching, after which 70 images were collected and quantified at 2 - s intervals. The fluorescence signals were measured in 6 pavement cells and blue lines represent best fits to multiexponential functions. Error bars indicate SD.

(D) Representative micrographs of VisG-TUB2 (upper) and VisG-TUB2^{E69K} (lower) in pavement cells. The photobleached area is shown by yellow boxes. Scale bars indicate 5 μm.

3) I still have trouble understanding exactly how the Glu69 site in TUB2 can account for sensing of cell wall integrity and subsequent signaling. The authors suggest that the exchange from a negatively charged amino acid to a positively charged amino acid can affect the interaction of TUB2 with Mg²⁺ and therefore affect microtubule polymerization and GTP hydrolysis. There are other possibilities that need to be

mentioned. For example, post-translation modifications (PTMs) - glutamate can be (poly)glutamylated, whereas lysine cannot. In fact, polyglutamylation is a common PTM of tubulin. On other hand, lysine can undergo PTMs that glutamate cannot (e.g., acetylation).

Response: Thank you for your insightful comments on the potential role of post-translational modifications (PTMs) at the 69th residue of TUB2. Polyglutamylation and acetylation are both well-characterized tubulin PTMs. In α -tubulin and β -tubulin, polyglutamylation occurs on several sites within the carboxy-terminal tails (Magiera & Janke, 2014) (residues 429–450 in Arabidopsis TUB2), making it unlikely that Glu69 undergoes polyglutamylation at this position. While the acetylation modifications have been detected on the amino acids at luminal surface of microtubules (Chu *et al*, 2011; Sadoul & Khochbin, 2016), suggesting that Lys69 may represent a potential site for acetylation. We have added a new paragraph to discuss this possibility in detail. Please see lines 285-298.

Cortical microtubules are highly dynamic, with the ability to self-organize aligned to the direction of maximal tensile stress in plant cells (Yan *et al*, 2023). Thus, cortical microtubules have long been considered as the sensor to respond to mechanical stress from cell wall (Hamant *et al*, 2019). Intriguingly, prior work has shown that the microtubule response to mechanical stress functions as an autonomous pathway, which is independent of FERONIA-mediated wall integrity maintenance (Malivert *et al*, 2021) (Please see Responsive Figure 3). Thus, we speculate that in the FRA1-deficient background, the mechanical stress caused by the non-cellulosic defects may activate the cell wall integrity signaling, which depends on normal microtubule function, and this activation is likely disrupted by the incorporation of TUB2^{E69K} into microtubules. Future studies need to be conducted to explore the precise molecular mechanisms by which Glu69 senses cell wall integrity and initiates downstream signaling. Thus, in the revised manuscript, we have explicitly described the limitations of our study and highlighted the key questions that require further investigation in the future. Please see lines 319-341.

REDACTED: Figure 5D and related text from Maliert A et al, 2021 (<https://doi.org/10.1371/journal.pbio.3001454>)

Minor points:

1) Lines 166 and 167 - I understand what the authors mean, but it is unclear from the text which hypothesis the authors are testing here. Please rewrite.

Response: Thanks for pointing this out. We have corrected “To further confirm this hypothesis” to “To confirm that the E69K mutation of TUB2 is a gain-of-function mutation rather than a null mutation”. Please see lines 135-137.

2) Lines 227 to 228 (and Figure 3F) - The oryzalin concentrations are quite high, in the micromolar range. In my experience, an application of oryzalin in the nanomolar range is enough to depolymerize microtubules. Can the authors confirm this is the correct concentration, and perhaps explain why such high concentrations were used? In the other experiments where they look at root swelling (Figure S3), the authors also use oryzalin in nanomolar concentrations.

Response: Thank you for your careful review and insightful comments. We used nanomolar concentrations of oryzalin in our root swelling experiments (Figure EV3), as this was sufficient to induce the swollen root phenotypes. In contrast, in Figure 2G (Figure 3F in the original manuscript), 6-day-old seedlings with an emerging pair of primary leaves were transferred to ½ MS solid medium containing oryzalin for 24 hours and the abaxial epidermal cells of the central vein in primary leaves were chosen to examine the fluorescent signal of microtubules. We have tested a series of concentrations and found that micromolar levels of oryzalin can effectively depolymerize microtubules under our experimental conditions. This method of application (the addition of oryzalin to the solid medium) and the tissue-specific differences in oryzalin sensitivity (abaxial epidermal cells) may contribute to the need for higher concentrations.

We agree that in many cases, nanomolar concentrations of oryzalin are sufficient to depolymerize microtubules, but this depends on multiple variables, including application method (liquid immersion / localized application / solid medium supplementation), incubation duration, plant growth stage, and the tissue or cell type being analyzed. There are also some published studies where micromolar concentrations of oryzalin have been used to study microtubule dynamics in leaf cells (Iida *et al*, 2023; Inaba *et al*, 2023). To avoid any confusion, we have provided a detailed description of the oryzalin treatments in the Materials and Methods section. Please see lines 452-458.

3) Figure 3F - The authors should provide a quantification of the disappearance of microtubules upon treatment with oryzalin for 24 hours in TUB2 in comparison to TUB2E69K plants. They state 5 biological replicates were tested, but no other numbers are shown.

Response: Thank you for your careful review and constructive feedback. We have quantified the microtubule density after the oryzalin treatment and added the quantification results to **Figure 2H**. The statistical methods are also described

in the Methods and Protocols section. Please see lines 440-443.

4) Figure 4B - I have trouble seeing the lateral displacement the authors describe. Are the yellow dots covering the displaced microtubule? If yes, this microtubule should be shown in a different way. Alternatively, a video could be uploaded to show this event.

Response: Thank you for pointing this out. In the Responsive Figure 3 below, the top panel highlights the displaced microtubule with yellow dots and the middle panel contains no such markings (Figure 4B in the original manuscript). We sincerely apologize for the visualization confusion caused by the dense microtubule networks in the selected region. To address this concern, we have replaced it with a new region featuring a clearer displaced microtubule, which can be seen in the bottom panel of Responsive Figure 4 or Figure 3B in the revised manuscript. We also uploaded a video to show the dynamic lateral displacement of microtubules in the *sofa1* mutant (Please see Movie EV1).

Figure 4B in the original manuscript

Figure 3B in the revised manuscript

Responsive Figure 4. Example of the lateral displacement of cortical microtubules labeled by VisGreen-TUB6 in *sofa1* mutant.

Top panel highlights the displaced microtubule with yellow dots and the middle panel contains no such markings in Figure 4B of the original manuscript. Bottom panel shows a new region featuring a clearer displaced microtubule in Figure 3B of the revised manuscript. Scale bars indicate 2 μ m.

5) Figure 4C - It looks like some lateral displacement events in the WT were not assigned with yellow arrowheads. The authors should mark these events appropriately not to mislead the reader.

Response: Thank you for your careful review and constructive feedback. In kymographs, microtubules that stably maintain their position appear as vertical lines, whereas the laterally drifting microtubules appear as slanted lines. We examined the selected regions that generated the WT kymograph shown in the original Figure 4B, and found there were no clear lateral displacement events. The slightly slanted

lines may be caused by the focus shifting of images. Based on the results of frequency of displaced MT (Figure 3E in the revised manuscript), we have generated a more representative kymograph from WT cells (Please see Figure 3C in the revised manuscript).

6) Grammar/spelling should be checked throughout the manuscript (for example line 388 'defeciency' instead of 'deficiency')

Response: Thank you for pointing this out. We have carefully checked the full text and corrected the spelling.

Referee #2:

The manuscript is very interesting; the discovery of TUB2 E69K mutation and its ability to restore the growth defect of fra1 mutant will advance the knowledge on microtubule dynamics and cell wall integrity signalling. The authors provided solid genetic evidence, but the regulatory mechanisms were not fully explored. I understand that studies published in EMBO Reports may prefer novelty and functional insight over extensive mechanistic detail, but still, I believe adding some additional evidence on cell wall integrity signalling or more insightful discussion will be helpful. Please see my specific comments below:

Response: We sincerely appreciate your thorough review of our manuscript and the positive feedback on its novelty and potential advances in understanding microtubule dynamics and cell wall integrity signaling.

Major issue.

1. The idea that TUB2E69K blocks the regulation of CWI gene expression in fra1 mutant is compelling; however, this conclusion is only supported by the RNA-seq data. If possible, the author may consider KO some key genes in the CWI signalling pathway; could this partially rescue the fra1 mutant phenotype? Alternatively, some chemical or stress treatment will also be helpful (I don't have any ideas in mind; treat the plant with something that relies on CWI-signaling, but the sofa mutant is less sensitive?)

Response: Thanks for your constructive suggestions. Regarding the knockout of key CWI signaling genes, it's unfortunately very difficult to implement. The CWI signaling pathway involves multiple genes that potentially act in a coordinated manner, making it challenging to achieve a meaningful phenotypic rescue by knocking out individual genes. In the future, we will focus our research on elucidating the key downstream genes involved.

Regarding the suggestion to use chemical or stress treatments to test CWI signaling, we agree that this is a valuable approach to strengthen our findings. The reported

chemical or stress treatment of CWI signaling mainly focuses on several plasma membrane receptors, including best-characterized FERONIA (FER) and THESEUS1 (THE1). THE1-mediated CWI signaling is activated by the isoxaben treatment (mimicking cellulose biosynthesis defects), which induces the accumulation of ectopic lignin (Hematy *et al*, 2007). FER-dependent signaling maintains cell wall integrity during salt stress (Feng *et al*, 2018) and brassinosteroid (BR)-induced cell expansion (Chaudhary *et al*, 2025). Referring to these studies, we treated WT, *fra1*, and *sofa1* seedlings with isoxaben, salt stress and brassinolide (BL, the most active form of BR), and found that there were no significant differences in the growth response among these genotypes (Please see Responsive Figure 5). Importantly, these results are consistent with the notion that the microtubule response to mechanical stress functions as an autonomous pathway, which is independent of FERONIA-mediated wall integrity maintenance (Malivert *et al*, 2021)

Given the existence of multiple independent cell wall integrity signaling pathways, establishing an integrated microtubule-mediated CWI signaling network and directly identifying the involved stress treatments would be extremely challenging. Thus, in the revised manuscript, we have explicitly described the limitations of our study and highlighted the key questions that require further investigation in the future. Please see lines 319-341. We prefer to propose such a possibility that microtubules function in the cell wall integrity induced by *FRA1*-deficiency, which potentially provides a foundation for subsequent studies to investigate the microtubule-mediated cell wall integrity signaling pathway.

Responsive Figure 5. The stress treatments related to CWI signaling on WT, *fra1* and *sofa1* plants.

(A) Phloroglucinol staining showed the lignin accumulation in root tip of WT, *fra1* and *sofa1* plants. The 5-day-old seedlings were incubated with mock (DMSO) or 600 nM isoxaben for 16 hours. Scale bars indicate 200 μ m.

(B) Quantitation of the primary root length under a range of BL concentrations for 6 days among three genotypes. Error bars indicate SD. ns ($p > 0.05$), no significant difference with WT by One-way ANOVA. 18 seedlings were counted for each sample.

(C) Quantitation of the root growth length of WT, *fra1* and *sofa1* seedlings. 6-day-old seedlings were transferred to $\frac{1}{2}$ MS solid medium containing a range of NaCl concentrations for 24 hours. ns ($p > 0.05$), no significant difference with WT by One-way ANOVA. 18 seedlings were counted for each sample.

Minor issues

1. Figure 3F requires some quantification of microtubule density.

Response: Thank you for your careful review and constructive feedback. We have quantified the microtubule density after the oryzalin treatment and added the quantification results to **Figure 2H**. The statistical methods are also described in the Methods and Protocols section. Please see lines 440-443.

2. Figure 4. The authors should provide additional evidence on the level of CUM1 in different genetic backgrounds. e.g qPCR to prove the expression of CUM is equal at the transcriptional level, or an immune-blot? The level of CUM should be identical in all backgrounds, but it might be miss-localized or degraded in the *fra1* or *sofa1* mutant? Also, it should be mentioned in the text as CUM1-labelled microtubules (lines 237-256).

Response: Thank you for your careful review and constructive feedback. We sincerely apologize for the confusion caused by the incomplete labeling in the manuscript. We used VisGreen-TUB6 to observe the lateral instability of microtubules in WT, *fra1* and *sofa1* plants, rather than CUM1-labeled microtubules (lines 237-256 in the original manuscript). We have revised the manuscript to clarify this point and avoid any confusion. Please see lines 216-217.

According to your suggestion, we performed qPCR analysis on WT, *fra1* and *sofa1* seedlings expressing mCherry-CMU1 to examine the transcriptional levels of CUM1 and found there was no significant difference among the three genotypes (Please see Responsive Figure 6A).

Additionally, we performed immunoblot analysis with an anti-mCherry antibody. The amount of mCherry-CMU1 protein in *sofa1* is similar to that in *fra1* mutant, both lower than WT plants (Please see Responsive Figure 6B). This result is consistent with the previous study that the changes in signal intensity of mCherry-CMU1 protein between *fra1* and WT plants are due to the alteration of protein levels (Ganguly *et al.*, 2020). Moreover, Ganguly *et al.* treated the *fra1* seedlings expressing mCherry-CMU1 with the 26S proteasome inhibitor MG132 and found the signal intensity restored to WT levels, and thus proposed that CMU1 is

degraded by the proteasome system and FRA1 regulates the extent of cortical microtubule localization of CMUs through protein degradation (Ganguly *et al*, 2020).

Responsive Figure 6. Transcriptional and protein levels of CMU1 in WT, *fra1* and *sofa1* plants.

(A) The relative expression of *CMU1* in WT, *fra1* and *sofa1* plants expressing mCherry-CMU1 detected by RT-qPCR. Data are presented as mean ± SD. ns ($p > 0.05$), no significant difference with WT by One-way ANOVA. $n=3$ biological replicates.

(B) Immunoblotting of mCherry-tagged CMU1 protein levels in three genotypes. Total protein was extracted from 10-day-old seedlings and was probed with anti-mCherry antibodies. Ponceau S staining of rubisco large subunit serve as loading control.

3. CMU1 was previously known as KLCR1 (kinesin light chain-related protein, Burstenbinder J Biol Chem 2013), but its name became slightly complicated when a group gave it a different name CMU for some reason (Liu et al., Dev Cell 2016). For good practice, both names should be used (or at least mentioned) in the manuscript (see Zang et al. Curr Biol, 2021). And some original papers related to KLCR/CMU should also be discussed.

Response: Many thanks for your valuable comments. We have corrected “cellulose synthase-microtubule uncoupling (CMU) proteins” to “cellulose synthase-microtubule uncoupling (CMU) / kinesin light chain-related (KLCR) proteins” upon their first mention in the revised manuscript (Please see lines 91-92). Meanwhile, we have discussed some original papers related to KLCR/CMU in lines 206-210. Thank you again for your attention to these important details.

Referee #3:

This manuscript by Yang and colleagues describes the isolation and characterization of the novel suppressor allele of the *fra1*/kinesin-4a mutant, *sofa1*. The authors show that *sofa1* suppresses the dwarf phenotype of *fra1*, but does not suppress the defects in cell wall mechanics and lateral microtubule stability of *fra1*. The authors also show

that the *sofa1* mutation in TUB has a dominant effect on the *fra1* dwarf phenotype. The authors further examined the cell wall composition and gene expression patterns in *fra1* and *sofa1*. Based on the results, the authors claim that the *sofa1* mutation blocks cell wall integrity signaling caused by transcriptional reprogramming induced by cell wall defects in *fra1*. I think the authors provide sufficient data to show that the E69K mutation in TUB has a dominant effect in suppressing the shoot growth phenotype of *fra1*. This must be an important finding to further our understanding of the function of Fra1. However, I think the evidence for cell wall integrity signaling and transcriptional reprogramming as described in lines 41-45, 341-345 and elsewhere is lacking.

Response: Thank you for your valuable feedback and for acknowledging the significance of our findings regarding the E69K mutation in TUB and its dominant effect on suppressing the shoot growth phenotype of *fra1*.

The authors' conclusion is based on the results that the expression levels of genes related to stress response and cell wall modulation were altered in *fra1* but not in *sofa1*. However, it is unclear whether these genes are essentially involved in the cell growth phenotype of *fra1* shoots or are indirect results of the developmental defects of *fra1*. That is, it is possible that the recovery of altered gene expression in *sofa1* is caused by restoration of cell growth rather than by blocking cell integrity signaling. The authors mention that down-regulation of XTH genes and EXPA14 accounts for the dwarf phenotype (lines 326-328). However, EXPA14 and some of the affected XTH genes are preferentially expressed in roots. Are there any reports that these genes are involved in cell elongation in shoots? Do any of the knockouts of these genes cause *fra1*-like shoot defects?

Response: We fully appreciate your concerns regarding the relationship between gene expression alterations and the shoot growth phenotype. These cell wall-related genes are down-regulated in the dwarf *fra1* mutant; however, when the E69K substitution occurs in TUB2 within the *fra1* mutant, plant height recovers to a WT-like level, and the expression of these wall genes also returns to a WT-like level. Although it remains uncertain whether these genes are directly responsible for the dwarf phenotype in *fra1*, their expression patterns suggest a potential association with the alteration of the plant height among the three genotypes. Thus, we have revised and weakened the statements into “the down-regulation of these cell wall genes may be associated with the dwarf phenotype in *fra1*”, rather than “the down-regulation of these cell wall genes accounts for the dwarf phenotype”. Please see lines 275-276 in the revised manuscript and the following response.

Although there is no direct evidence that knockout of these cell wall genes causes *fra1*-like shoot defects, some studies have reported their involvement in regulating shoot cell elongation. For example, *XTH8* gene has been reported to be involved in the regulation of salicylic acid (SA)-dependent dwarfism (Miura *et al*, 2010) (Please see Responsive Figure 7); *XTH15::GUS* is also expressed in the inflorescence stem (Becnel *et al*, 2006) (Please see Responsive Figure 8) and *xth15* mutants have reduced petiole growth rates under shade conditions (Sasidharan *et al*, 2010)

(Please see Responsive Figure 9); Recently, Balkova *et al.* have demonstrated the localization of EXPA14 in the cell wall of the shoot epidermis, suggesting its role in modulating shoot structure and function (Balkova *et al.*, 2025) (Please see Responsive Figure 10). These genes may be involved in the regulation of cell elongation in the *fra1* mutant in a coordinated manner. We have discussed these studies in the revised manuscript. Please see lines 305-318.

Based on the following results: 1) The TUB2^{E69K} does not disrupt the overall microtubule cortical array organization or the cellulose biosynthesis; 2) The TUB2^{E69K} restored plant height of *fra1* mutant without restoring the non-cellulosic deficiency; 3) The TUB2^{E69K} did not affect the plant height of WT plants; 4) The similar gene expression patterns between WT and *sofa1*, and the differential expression of a large number of stress and cell wall-related genes in the *fra1* mutant; 5) The potential role of microtubules in cell wall integrity signaling, we proposed that the E69K mutation in TUB2 potentially regulates cell elongation by blocking the cell wall integrity signaling in *FRA1*-deficiency background. We fully acknowledge that the precise molecular mechanisms remain unclear and require further investigation. We prefer to propose such a possibility that microtubules function in the cell wall integrity induced by *FRA1*-deficiency, which potentially provides a novel perspective for future research.

REDACTED: Figure 9 and related text from Miura K et al, 2010 (<https://doi.org/10.1093/pcp/pcp171>)

REDACTED: Figure 4 and related text from Becnel J et al, 2006 (DOI 10.1007/s11103-006-0021-z)

REDACTED: Figure 9 and related text from Sasidharan R et al 2010 (<https://doi.org/10.1104/pp.110.162057>)

REDACTED: Figure 1E-J and related text from Balkova D et al 2025 (<https://doi.org/10.3389/fpls.2025.1546819>)

I think there is a gap between the presented data and the authors' proposal that cortical microtubules (or TUB?) serve as a sensor of cell wall integrity, and the model presented in Figure 7 and in the abstract is not supported by the presented study. I suggest that the authors weaken their claim on cell wall integrity signaling as one of the possible explanations and provide more reasonable model(s) without jumping to an unsupported conclusion.

Response: Thank you for your constructive feedback. We acknowledge that our previous conclusions that cortical microtubules serve as a sensor of cell wall integrity were speculative. In light of your comments, we have made comprehensive revisions to weaken the potential role of microtubules as a sensor for cell wall damage in our conclusion, and focused more on the possibility that TUB2^{E69K} may block the CWI signaling pathway to alleviate the growth inhibition. The details are as follows:

1) We have removed the conclusion that “cortical microtubules serve as the sensor of cell wall integrity” in the abstract and introduction section. Please see lines 37-44 and 98-102;

2) We have removed the description of "transcriptional reprogramming", and emphasized that these differentially expressed genes among the three genotypes may be potentially associated with cell elongation.

3) We have proposed that cell wall integrity signaling as one possible explanation

rather than a definitive conclusion. Please see lines 319-325;

4) We have also removed the extensive discussion of cell wall integrity signaling and explicitly described the limitations of our study. Please see lines 326-331;

5) We have explored other potential explanations for the observed phenotypes in the revised manuscript. Please see lines 331-341;

6) We have removed the model diagram originally presented in Figure 7 in the revised manuscript;

7) Based on the above-mentioned modifications, we re-formatted the manuscript as a Scientific Report.

We thank you for your valuable suggestions and hope that these revisions address your concerns and better reflect the scope and findings of our study.

Minor points.

1. Line 233-235. This is limited to the *fra1* mutant. To prove this, the authors should examine MT dynamics and/or oryzalin sensitivity of TUB(E69K)-induced cells in wild type background, not in *fra1* background.

Response: Thank you for your careful review. We have also conducted oryzalin treatment on both WT plants and *tub2*^{E69K} point mutants. Similar to Figure EV3, the *tub2*^{E69K} mutants are more sensitive than WT plants (Please see Responsive Figure 11), indicating that the incorporation of TUB2^{E69K} affects the stability of microtubules, in either the wild type or *FRA1*-deficient background. Since our study primarily focuses on the suppression of the *fra1* dwarf phenotype, we have revised the manuscript to explicitly state that the E69K mutation in TUB2 affects microtubule stability in the *fra1* mutant background. Please see lines 201-202.

Responsive Figure 11. Root phenotypes of WT and *tub2*^{E69K} plants treated with oryzalin.

(A) Primary roots of 6-day-old WT and *tub2*^{E69K} seedlings exposed for 24 hours to various concentrations of oryzalin. Scale bars indicate 100 μm.

(B, C) Quantification of the root length (B) and root width (C) as shown in (A). Data are presented as mean \pm SD. n=8 biological replicates.

2. Figure legends are incomplete (Figures 2-5).

Response: Thank you for pointing this out. We have revised and supplemented all figure legends to provide more complete and detailed descriptions. Additional information can also be found in the Methods and Protocols section.

3. line 349-351, incomplete sentence?

Response: We apologize for the oversight. We have corrected it and carefully checked through the full text to ensure completeness and clarity. Thank you for your attention to detail.

Cited reference in this point-by-point response file:

Balkova D, Mala K, Hejatko J, Panzarova K, Abdelhakim L, Pleskacova B, Samalova M (2025) Differential expression and localization of expansins in shoots: implications for cell wall dynamics and drought tolerance. *Front Plant Sci* 16:1546819

Becnel J, Natarajan M, Kipp A, Braam J (2006) Developmental expression patterns of Arabidopsis genes reported by transgenes and Genevestigator. *Plant Mol Biol* 61: 451-467

Chaudhary A, Hsiao YC, Jessica Yeh FL, Zupunski M, Zhang H, Aizezi Y, Malkovskiy A, Grossmann G, Wu HM, Cheung AY *et al* (2025) FERONIA signaling maintains cell wall integrity during brassinosteroid-induced cell expansion in Arabidopsis. *Mol Plant* 18: 603-618

Chu CW, Hou F, Zhang J, Phu L, Loktev AV, Kirkpatrick DS, Jackson PK, Zhao Y, Zou H (2011) A novel acetylation of beta-tubulin by San modulates microtubule

polymerization via down-regulating tubulin incorporation. *Mol Biol Cell* 22: 448-456

Feng W, Kita D, Peaucelle A, Cartwright HN, Doan V, Duan QH, Liu MC, Maman J, Steinhorst L, Schmitz-Thom I *et al* (2018) The FERONIA Receptor Kinase Maintains Cell-Wall Integrity during Salt Stress through Ca Signaling. *Current Biology* 28: 666-675

Ganguly A, Zhu CM, Chen WZ, Dixit R (2020) FRA1 Kinesin Modulates the Lateral Stability of Cortical Microtubules through Cellulose Synthase-Microtubule Uncoupling Proteins. *Plant Cell* 32: 2508-2524

Hamant O, Inoue D, Bouchez D, Dumais J, Mjolsness E (2019) Are microtubules tension sensors? *Nat Commun* 10: 2360

Hematy K, Sado PE, Van Tuinen A, Rochange S, Desnos T, Balzergue S, Pelletier S, Renou JP, Hofte H (2007) A receptor-like kinase mediates the response of Arabidopsis cells to the inhibition of cellulose synthesis. *Curr Biol* 17: 922-931

Hush JM, Wadsworth P, Callaham DA, Hepler PK (1994) Quantification of microtubule dynamics in living plant cells using fluorescence redistribution after photobleaching. *J Cell Sci* 107 (Pt 4): 775-784

Iida H, Mahonen AP, Jurgens G, Takada S (2023) Epidermal injury-induced derepression of key regulator ATML1 in newly exposed cells elicits epidermis regeneration. *Nat Commun* 14: 1031

Inaba H, Oikawa K, Ishikawa K, Kodama Y, Matsuura K, Numata K (2023) Binding of Tau-derived peptide-fused GFP to plant microtubules in Arabidopsis thaliana. *PLoS One* 18: e0286421

Magiera MM, Janke C (2014) Post-translational modifications of tubulin. *Curr Biol* 24: R351-354

Malivert A, Erguvan O, Chevallier A, Dehem A, Friaud R, Liu M, Martin M, Peyraud T, Hamant O, Verger S (2021) FERONIA and microtubules independently contribute to mechanical integrity in the Arabidopsis shoot. *PLoS Biol* 19: e3001454

Miura K, Lee J, Miura T, Hasegawa PM (2010) Controls Cell Growth and Plant Development in Arabidopsis Through Salicylic Acid. *Plant Cell Physiol* 51: 103-113

- Sadoul K, Khochbin S (2016) The growing landscape of tubulin acetylation: lysine 40 and many more. *Biochem J* 473: 1859-1868
- Sasidharan R, Chinnappa CC, Staal M, Elzenga JTM, Yokoyama R, Nishitani K, Voesenek LACJ, Pierik R (2010) Light Quality-Mediated Petiole Elongation in Arabidopsis during Shade Avoidance Involves Cell Wall Modification by Xyloglucan Endotransglucosylase/Hydrolases. *Plant Physiol* 154: 978-990
- Shaw SL, Kamyar R, Ehrhardt DW (2003) Sustained microtubule treadmilling in Arabidopsis cortical arrays. *Science* 300: 1715-1718
- Vorobjev IA, Rodionov VI, Maly IV, Borisy GG (1999) Contribution of plus and minus end pathways to microtubule turnover. *J Cell Sci* 112 (Pt 14): 2277-2289
- Yan Y, Sun Z, Yan P, Wang T, Zhang Y (2023) Mechanical regulation of cortical microtubules in plant cells. *New Phytol* 239: 1609-1621

Dear Zhaosheng,

Thank you for submitting your revised manuscript. It has now been seen by all original referees. My apologies for the delay in getting back to you, which was due to the delay in receiving referee reports and conference travel.

As you can see, referees find that the study is significantly improved during revision and recommend publication. However, I need you to address the points below before I can accept the manuscript.

- Please note that as per EMBO Press policy all source data underlying figure panels must be published alongside the paper. We note that source data for 1D has not been provided. Please provide source data for Figure 1D.
- Please make sure that the funding information is also complete in the manuscript tracking system, which is currently missing - i.e. grants from the State Key Laboratory of Plant Genomics.
- We note that Figure EV4 is currently not called out in the manuscript text.
- Our production/data editors have asked you to clarify several points in the figure legends - Figure Legends (main + EV):
 - o Please note that the exact p values are not provided in the legends of figures 1B, H; 2B, C, H; 3D, E; 4A, EV1 A, C, D, F, I; EV2 C
 - o Please note that the measure of center for the error bars needs to be defined in the legends of figures 1B, H; 2B, C, D, E, F, H; 3D, E; 4A, EV1 A, C, D, F, I; EV2 C, E.

Thank you again for giving us to consider your manuscript for EMBO Reports, I look forward to your minor revision.

Kind regards,

Deniz

--

Deniz Senyilmaz Tiebe, PhD
Senior Scientific Editor
EMBO Reports

Referee #1:

The authors have addressed my major concerns, and have made changes to the manuscript in accordance with these recommendations. I have no further comments or concerns.

Referee #2:

I'm happy with the revision.

Referee #3:

I appreciate the authors' thorough response to my feedback. I believe the authors revised the manuscript properly. They effectively weakened their claim on the mechanism by which the TUB mutation suppresses the fra1 phenotype, thus aligning it with the level of support provided by the current evidence.

The authors have addressed all minor editorial requests.

Prof. Zhaosheng Kong
Institute of Microbiology, Chinese Academy of Sciences
State Key Laboratory of Plant Genomics
1 Beichen West Road
Chaoyang District, Beijing 100101
China

Dear Zhaosheng,

Thank you for submitting your revised manuscript. I have now looked at everything and all is fine. Therefore, I am very pleased to accept your manuscript for publication in EMBO Reports.

Congratulations on a nice work!

I need your input on one more point before we can export your manuscript to our publishers. To increase clarity and accessibility of the findings, I would like to suggest the below minor change in the title. Please take a look and confirm, or feel free to propose further changes:

E69K mutation in β -Tubulin 2 blocks cell wall integrity signaling during plant cell elongation

Kind regards,

Deniz

--

Deniz Senyilmaz Tiebe, PhD
Senior Scientific Editor
EMBO Reports

Yours sincerely,

Deniz Senyilmaz Tiebe, PhD
Senior Scientific Editor
EMBO Reports
